# Beyond the words: Exploring individual differences in the evaluation of poetic creativity

**Soma Chaudhuri**[1]*, **Alan Pickering**[1], **Maura Dooley**[2], **Joydeep Bhattacharya**[1]

**1** Department of Psychology, Goldsmiths, University of London, London, United Kingdom, **2** Department of English and Creative Writing, Goldsmiths, University of London, London, United Kingdom

* schau002@gold.ac.uk

## Abstract

Poetry is arguably the most creative expression of language and can evoke diverse subjective experiences, such as emotions and aesthetic responses, subsequently influencing the subjective judgment of the creativity of poem. This study investigated how certain personality traits—specifically openness, intellect, awe-proneness, and epistemic curiosity–influence the relationship between these subjective experiences and the creativity judgment of 36 English language poems. One hundred and twenty-nine participants rated each poem across six dimensions: clarity, aesthetic appeal, felt valence, felt arousal, surprise, and overall creativity. Initially, we obtained a parsimonious model that suggested aesthetic appeal, felt valence, and surprise as key predictors of poetic creativity. Subsequently, using multi-level analysis, we investigated the interactions between the four personality traits and these three predictors. Among the personality traits, openness emerged as the primary moderator in predicting judgments of poetic creativity, followed by curiosity and awe-proneness. Among the predictors, aesthetic appeal was moderated by all four personality traits, while surprise was moderated by openness, awe-proneness, and curiosity. Valence, on the other hand, was moderated by openness only. These findings provide novel insights into the ways individual differences influence evaluations of poetic creativity.

**Data Availability Statement:** The Open Science Framework (OSF): https://osf.io/9mw7r/?view_only=07137f4871d146c790501f22bc7743d5.

**Funding:** The author(s) received no specific funding for this work.

## Introduction

Poetry, one of the most creative forms of linguistic expression used since ancient times, served as a powerful medium to communicate emotions, thoughts, and ideas [1–3]. However, despite its unique status in human culture, how we evaluate the creativity of poems remain underexplored. This gap may stem from the inherent subjectivity that characterizes poetry as a literary art form. The essence of a poem's impact lies in its ability to connect with readers on a deeply personal level; we appreciate poetry for how well it engages our thoughts and feelings [4]. The adage "Beauty is in the eye of the beholder,"[5] aptly captures the subjective nature of aesthetic appreciation, a principle that applies equally to poetry. The creative value assigned to a poem can vary widely among individuals, influenced by their subjective experiences. Readers

**Competing interests:** The authors have declared that no competing interests exist.

comprehend the same poem differently depending on their knowledge and perceptual ability introducing a degree of variability in evaluating a poem's creativity. What one individual might find creative and captivating, another may find ordinary or unappealing. Such variability can be attributed to the differences in personality traits of readers, which are likely to influence their assessments, and subsequently, their overall creativity judgment of poetry. This study investigated how readers' internal models formed by their personality traits impact their subjective feelings and experiences of reading poems while assessing poetic creativity.

The 4P model of creativity, a seminal theoretical framework of creativity, proposed "The word creativity is a noun naming the phenomenon in which a person communicates a new concept (which is the product). Mental activity (or mental process) is implicit in the definition and of course no one could conceive of a person living or operating in a vacuum, so the term press is also implicit. The definition begs the questions as to how new the concept must be and to whom it must be new" [6]. Among these 4P approaches, i.e., person, product, process, and press, the product or physical object, plays an important role. In common perceptions, creativity is often equated with its tangible outcome—the creative product. When asked to define creativity, many would instinctively describe it in terms of the final product [7]. Literature suggests that a product-centered operational definition is the most useful for empirical research in creativity and presumably the most important feature of this definition is its reliance on subjective criteria [8]. Despite debates and the difficulty of precisely defining creativity of a product [9–11], the most widely accepted operational definition is the "standard definition" of creativity, which states that for a product or idea to be deemed creative, it must be both original or novel and useful or appropriate. Additionally, surprise is also added as the third ingredient of creativity [12]. The process aspect of the 4P model usually involves two phases of cognitive processes: the generative phase and the evaluative phase [13].

The present study adopts a dual focus on both the product and process aspects of creativity using poem as the product and its evaluation process as the measure of creativity. We operationalized the 'creativity' of a poem as its creative potential, aiming to broaden the understanding of creativity from the creator to the creation itself. Our approach is in line with past studies that have investigated the creativity evaluation of various types of products/artefacts, such as ideas [14], musical compositions [15,16], short stories [17], and product concepts [18], to name a few. This approach allows us to investigate how individuals assess the creativity of poems, recognizing the subjective nature of such evaluations and how they may be influenced by individual personality traits. In summary, we aim to uncover how variations in reader personality may subtly influence the evaluation of a poem's creativity, thereby shaping an implicit model of evaluation. When assessing the creativity of a product, raters often form their own mental criteria, which can vary depending on their knowledge, personal preferences and personality traits [19]. Personality traits are basic dimensions on which people differ, reflecting their characteristic patterns of thoughts, feelings, and behaviours with consistency and stability [20,21]. Several studies [22–25] have investigated the link between personality traits and creativity. Significant positive correlations have been observed between different measures of creativity and Big Five personality traits [26–28], especially with openness to experience [29–31]. A meta-analysis [23] identified openness to experience as the predominant personality trait consistently positively correlated with the creative potential of individuals in both the Arts and Sciences. Research also suggests that openness to experience is positively correlated with rater discernment ability to distinguish creative from uncreative responses—open people do not merely rate all responses as more creative rather, they are better at identifying genuinely creative ideas, thereby demonstrating higher overall discernment [32,33]. Another recent study highlights how an individual's consideration of the novelty and usefulness of creativity task responses is influenced by contextual factors and individual differences, such as openness and

intellect, in overall creativity judgment [14]. Additionally, positive emotions, such as curiosity —defined as the desire to know [34,35]—have consistently demonstrated a significant correlation with creativity across multiple studies, as evidenced by their weighted effect sizes [36]. Awe, another positive emotion, has been linked to creative thinking [37]. These studies focused primarily on the relationship between personality traits and various creative idea-generation processes, such as divergent thinking, everyday creative behaviour, creative achievement, and self-rated creativity. However, the influence of personality traits on the evaluation of creativity of poetry has not been adequately explored. Of note, some studies have found that individual differences in visual imagery abilities, ambiguity tolerance, awe-proneness, and nostalgia-proneness predict the aesthetic appeal of specific forms of poems like haiku and sonnets [38–40].

In this study, consistent with prior research, we focused on four personality traits among readers: openness, intellect, awe-proneness, and epistemic curiosity. We aimed to explore how these traits influence the assessment of poem creativity. Initially, we identified predictors for assessing the creativity of an English poem. Following prior research [8,41–43], we selected five potential predictors: clarity, aesthetic appeal, felt valence, arousal, and surprise. Subsequently, we examined how the selected personality traits might moderate the influence of the predictors on the creativity judgment of a poem. In the following sections, we provide a brief overview of these potential predictors, the personality traits under consideration, and their prospective roles in evaluating creativity.

## Clarity

Clarity in a text means it is lucid, understandable, and comprehensible to the readers. This quality is especially valuable in written communication forms like poetry, where the goal is for readers to grasp the intended message. Previous research supports that clarity is an important factor in assessing the creativity of a poem [8].

## Aesthetic appeal

Aesthetic appeal refers to the artistic features, styles, and concepts present in any form of artwork. Research on the psychology of creativity and aesthetics has engaged with a variety of stimuli, including paintings and visual art [44–48], music [16,49–54], films [55–57], and poems [38–40,58–62]. Previous empirical studies on poetry have primarily investigated aesthetic appreciation focusing on two broad aspects: (i) the objective properties of a poem and (ii) the subjective experiences the poem evokes in readers. The first approach examines textual elements, e.g., rhythm, rhyme, meter [59,63], metaphors [58,64–66], and phonological constructs such as words and phrases [67,68]. The second approach explores empathic reactions and emotional involvement [69], perceived emotional valence and vividness in imagery [38], cognitive and emotional ambiguity (e.g., awe and nostalgia) [61,39], openness to experience, visual imagery abilities, felt valence [39], expertise [70], gender and ethnicity [71]. However, the potential interactions between these two approaches and how readers' characteristics influence their subjective evaluation of creativity remain unclear.

## Felt emotions

Felt valence describes the emotional tone experienced by the perceiver, indicating whether the emotion is positive or negative, whereas arousal refers to the intensity or strength of the emotional state felt. The two-dimensional circumplex model of emotion, proposed by Russell [72], conceptualizes emotional states along two orthogonal dimensions: valence (pleasure-displeasure: horizontal axis) and arousal (arousal-sleep: vertical axis). Poetry is known to evoke strong

emotional experiences [73] and these emotional states can influence creativity evaluation [74]. A recent study suggests that the content and prosodic features of poetry can evoke basic emotions, while a reader's intellectual evaluation of a poem can evoke a complex aesthetic emotion that combines a basic emotion with their assessment of the poem [75]. It is important to note in this context that perceived and felt emotions may be different. Research in music has consistently reported that perception of emotion involves sensory and cognitive processes that do not necessarily mirror the actual feelings of the perceiver. Hence, the emotion perceived or expressed by stimuli and the emotion felt by the perceiver may differ [76–78]. In our study, we focused on the felt emotions, i.e., the emotions felt by the reader while reading the poem, rather than the perceived emotion, i.e., the emotions expressed by the poem. Felt valence here reveals the extent to which the readers felt positive or negative emotions while reading the poems, whereas felt arousal reveals how intense it was felt by the readers.

## Surprise

Surprise is usually a short-lived emotion elicited by events that deviate from an established schema or expectations [79–81], where a schema refers to a component of the organism's knowledge structure, activated by a specific stimulus [82]. Surprise is recognized as a key predictor of the creativity of a product or idea [12,43], and is also a robust predictor of the aesthetic judgment of artwork [83]. As surprise describes the reaction to unexpectedness [80,84], in our study, we defined surprise as the extent to which the readers experienced a sudden and unexpected change in the context or theme of the poem.

## Openness and intellect

Openness to experience is a broad range of traits, from intellectual abilities to aesthetic and artistic interests [85–87], and is most robustly associated with measures of creativity [88]. It influences a variety of domains, including vivid fantasy [89], artistic sensitivity, novelty in artworks, aesthetic emotions [90], intellectual curiosity [91], and unconventional attitudes [88]. Openness and intellect, though characterized as a unified dimension of personality, can be differentiated into two major aspects: openness and intellect [92,93]. Based on different styles of cognitive exploration, openness reflects the tendency to engage with aesthetic and sensory information, both in perception and imagination. On the other hand, intellect is a dispositional individual difference variable related to intellectual performance, such as problem-solving, thinking, information search, learning, or creativity [85,94]. Further, openness has been identified as a predictor of creative accomplishments in the arts, whereas intellect predicts creative achievements in the sciences [27]. Therefore, we expected that openness and intellect would separately impact the relationship between aesthetic appeal and creativity ratings of a poem. Research consistently demonstrates that individuals with higher levels of openness are drawn to art in general and exhibit greater appreciation for unconventional artistic expressions [87,95,96]. Considering high openness as a characteristic of the "artistic personality"[87], we predicted that individuals with greater openness would prioritize aesthetic appeal while assessing creativity of a poem compared to those with lower level of openness. Cosidering intellect's link to abstract or semantic information, and acknowledging that underlying meaning or message conveyed through the words and language used in poetry contributes to its overall aesthetic quality, we expected individuals with higher intellect to prioritize aesthetic appeal while assessing poetic creativity.

Individuals with higher openness are known to be more sensitive and attuned to their feelings [97], yet intense emotional engagement can sometimes inhibit higher cognitive functions in these individuals [98]. Neurological studies suggest that heightened emotional states can

inhibit the brain's reflective processes, affecting intellectual openness [98]; see also [99]. Hence, we expected that openness would moderate the relationship between felt emotions [both valence and arousal] and creativity. Specifically, the positive impact of felt emotions on creativity ratings may be perceived as less pronounced by individuals with higher levels of openness compared to those with lower levels of openness. Considering intellect's link to complex information processing [26,100], we expected that intellect would not moderate the relationship between felt emotions and creativity evaluations, suggesting that the influence of emotions on creativity judgments would remain consistent regardless of individuals' levels of intellect.

Surprise, often triggered by unexpected or schema-discrepant events, requires significant cognitive engagement to assess violations of expectancy in poetry [79,80,101]. We predicted that both openness and intellect would moderate the relationship between surprise and creativity. Specifically, we expected that individuals high in open-mindedness and intellectual curiosity would exhibit a heightened receptivity and interest in unexpected elements within poems. This inclination would lead them to prioritize surprise when assessing the creativity of poems, in contrast to those with lower levels of openness and intellect.

## Awe-proneness

Awe, classified as an epistemic emotion, is a distinct emotional response to encountering something vast, both literally and figuratively, and requires cognitive accommodation [102]. Poetry is likely to elicit awe due to its rich information content [103]. Dispositional awe-proneness is significantly correlated ($r = 0.49$) with openness to experience [103]. Further, higher dispositional awe has been positively associated with aesthetic engagement and a tendency to experience aesthetic chills [104], which are transient emotional responses to aesthetical stimuli, manifesting as chills or waves of excitement when engaging with poetry or art [105]. Since awe is linked to surprise and amazement and is interpreted as a passive, receptive mode of attention in response to the unexpected [102], we predicted that the dispositional awe-proneness would moderate the effect of aesthetic appeal and surprise on a poem's creativity scores. Specifically, we predicted that the impact of aesthetic appeal and surprise on creativity ratings would be more pronouunced in individuals with higher levels of awe-proneness, who, due to their disposition, are more open and responsive to a poem's aesthetic qualities and unexpected elements, leading them to attribute higher creativity to such poems.

## Epistemic curiosity

Curiosity is a motivating positive emotion [106] and an intense desire to explore novel, complex and uncertain events [107]. It is associated with learning and thinking processes and linked to various constructs such as interest, surprise, confusion, and awe [108,109]. Curiosity can be categorized into two broad types: perceptual curiosity and epistemic curiosity; perceptual curiosity leads to increased perception of stimuli, and epistemic curiosity is defined as a "drive to know" [34]. Epistemic curiosity motivates individuals to engage in exploratory behaviours to bridge the gap between their existing knowledge and their desire for further understanding [35,110,111]. Also, highly open individuals tend to be curious about the world [112–115]. Therefore, we predicted that epistemic curiosity would significantly moderate the relationship between aesthetic appeal, surprise, and creativity. Specifically, we predicted that the positive impact of aesthetic appeal and surprise on creativity scores would be more pronounced in individuals with higher levels of epistemic curiosity. These individuals, driven by their curiosity, would be more inclined to appreciate the aesthetic qualities and unexpected elements in a poem, thus attributing higher levels of creativity to such poems.

## Materials and methods

### Materials

Initially, we selected 108 English poems spanning various genres, themes, and periods from online resources, including the Poetry.org (http://www.poetry.org/), the Poetry Foundation (https://www.poetryfoundation.org/), and the Academy of American Poets (https://poets.org/). These poems were subsequently evaluated for their levels of "surprise" by M.D., a Professor of English and Creative Writing with domain-specific expertise, using a scale of 1 to 7, where 1 indicates "absolutely not surprising" and 7 indicates "absolutely surprising." Following this evaluation, we shortlisted 36 poems as the experimental stimuli for our study: 18 with low surprise ratings (4 or lower) and 18 with high surprise ratings (6 or above). The chosen poems varied in structures, contents, lines, and word count (mean number of lines = 11, SD = 3.24; mean word count = 71.25, SD = 28.99). To represent a broad spectrum of English poems, we consciously avoided limiting our selection to a particular genre or form, like haiku or sonnets as done in previous studies [38,39,116].

The selected stimuli are both lexically and semantically diverse. Lexical diversity (LD) of a text refers to its lexical richness, indicating the range and variety of vocabulary deployed in the text [117]. We calculated LD using the type-token ratio (TTR) method, which calculates the ratio of unique words (types) to the total word count (tokens) [118]. It ranges from 0 to 1, with a higher TTR indicating a greater lexical diversity. The mean (SD) lexical diversity across the poems is 0.77 (0.09), suggesting that, on average, about 77% of the words used in the poems are unique or different. Semantic diversity, on the other hand, refers to the range of contexts (i.e., semantic richness) in which words are used [119]. We calculated the semantic diversity using divergent semantic integration (DSI) (http://semdis.wlu.psu.edu/), which calculates the mean semantic distance between all word pairs in a poem. DSI varies from 0 to 1, with a higher score indicating a broader collection of divergent ideas. The average (SD) semantic diversity across the poems is 0.80 (0.03), indicating a high degree of semantic variety (see S1 Table in the Supplementary section for details).

### Participants

By using the G*Power software (v. 3.1.9.4), [120] we found that a minimum sample size of 92 was required to detect a medium effect size ($f^2$ = 0.15) in a multiple linear regression, assuming a significance level of 0.05 and a statistical power of 80%. By employing a multilevel model considering 92 cluster groups, assuming a small to medium effect size (Cohen's d) of 0.3, and considering 36 observations per cluster group, 'samplesize_mixed' function in R (https://strengejacke.github.io/sjstats/) determined that a total sample size of 965 observations was necessary, indicating a minimum requirement of 27 participants (965/36). The criteria we used are widely-used conventional figures when estimating sample sizes. We recruited 129 adult participants via Prolific®, a participant-recruiting platform. As the task lasted approximately one hour, we excluded 30 participants who exceeded a two-hour time limit. Additionally, three participants were eliminated from the analyses due to their identical responses on the subjective rating measures across the poems. Our final sample consisted of 96 participants resulting in a total of 3456 observations, ensuring sufficient statistical power for our study. Participants (N = 96, 32 males, 63 females, 1 preferred not to say; mean age = 31.94 years, SD = 13.09) were fluent in English (self-reported) and from a variety of educational backgrounds holding at least a bachelor's degree in any discipline.

Participants were briefed about the experimental procedure, which involved the assessment of a set of English poems on a 7-point Likert scale (1 = extremely low; 7 = extremely high)

across various constructs including clarity, aesthetic appeal, felt valence, arousal, surprise, and overall creativity. Additionally, participants were instructed to complete demographic and personality-related questions. We assured participants of the full confidentiality of their data, in compliance with the General Data Protection Regulation, and clarified that any published results would be non-identifiable. All participants provided informed consent (online) before data collection. Participants were compensated £7.50 per hour for their participation. The data collection period spanned from 27 January 2022 to 23 June 2022, and the data were accessed for research purposes only after this period. The study protocol was approved by the local Ethics Committee of the Department of Psychology, Goldsmiths University of London.

## Procedure

The experiment was created using Qualtrics®, and the link was disseminated through Prolific®, a platform for participant recruitment. Participants received a broad overview of the study and comprehensive instructions for ratings. In the beginning, a sample poem was provided to facilitate a clearer understanding of the process. Participants were given a minimum of 30 seconds to read each poem. Following this period, they were allowed to proceed to the rating task. They were asked to rate the poems on six dimensions in the following order: clarity, aesthetic appeal, felt valence, felt arousal, surprise, and creativity, using a 7-point Likert scale (1: "Extremely Low" and 7: "Extremely High"). There was no time limit imposed for rating the poems. Of note, the poems remained visible during the rating process. A brief demographic survey was conducted once 36 trials were finished. Finally, participants completed a set of questions on personality traits–Ten Item Personality Inventory (TIPI: [121]), openness/intellect [92], awe-proneness [103], and epistemic curiosity [110]. All personality questionnaires utilized a 7-point scale, with 1 representing "disagree strongly" and 7 representing "agree strongly". It took an hour on average to finish the whole experiment.

## Analysis

The primary aim of our study was to explore how four personality traits—openness, intellect, awe-proneness, and epistemic curiosity—moderate the impact of significant potential predictors on poetic creativity. First, we determined the significant predictors of the creativity of poems. To accomplish this task, five maximum likelihood linear mixed models (predictor models) were executed using the *lme4* package [122] in R (version 4.0.3). We employed the forward selection approach to incorporate variables into the predictor model. Starting with the variable showing the highest correlation with the outcome variable, i.e., creativity, we sequentially added other variables in descending order of their correlations with creativity. Hence, the sequence of inclusion for the predictor variables was as follows: aesthetic appeal, felt valence, surprise, arousal, and clarity. The analysis included the five potential predictors (group mean centered) as fixed effects, with creativity as the outcome variable, and participants as the grouping variable. Additionally, random effects intercepts for participants were incorporated in the analysis. The best model fit results identified the potential predictors of poetic creativity.

The overall data visualisation confirmed that the response variable follows a normal distribution, and there is no significant multicollinearity among the independent variables (Variance Inflation Factor < 3). Furthermore, the reliability of the measurement was established by assessing the internal consistency across items (Cronbach's alpha = 0.80; McDonald's Omega Total = 0.88; Omega H asymptotic = 0.71, Omega Hierarchical = 0.62) [123,124].

The dataset comprised 3456 responses and exhibited a common multilevel structure, with individual responses (Level-1) nested within participants (Level-2). The null model revealed

that a significant 54% of the variance was attributed to the grouping variable (participants), affirming the necessity of employing a linear mixed model to accommodate the hierarchical nature of the data, over standard regression models. Furthermore, the intraclass correlation coefficient (ICC = 0.28), signifying the Level-2 clustering, revealed a significant level of clustering in the data. This implies that the Level-1 dependent variable (creativity) was not independent of the Level-2 grouping variable (participants). Hence, the use of multilevel modeling was considered appropriate.

To accurately estimate the within-group effects, the predictors were centered within clusters (CWC) before entering the models [125]. Finally, we examined the impact of four personality traits (e.g., openness, intellect, awe-proneness, and epistemic curiosity) on potential predictors by conducting four separate linear mixed models (personality traits models). In these models, the personality traits and their interactions with the potential predictors were treated as fixed effects, with creativity as the response variable and participants as the grouping variable. To visualize the interaction effects of the moderators on the predictors, we followed the classical convention [126]. Specifically, we plotted the mean value of the moderator and one standard deviation above and below the mean, allowing us to observe how the moderator influences the relationship between the predictors and creativity. The original measurement scales were 7-point scales. Before entering the model, five potential predictors were centered within each subject (i.e., group mean-centered) to obtain a clear estimate of the within-group effect [125]. For the interaction plots, it is a standard practice to use a scale that reflects the original range of the variables rather than the centered range. Therefore, on the X-axis, the scales for the predictors (group mean centered) range from -7 to +7, while the outcome variable (uncentered) on the Y-axis ranges from 1 to 7.

## Results

### Descriptive statistics

Descriptive statistics of the variables related to ratings on poems and personality trait scores of participants are shown in Table 1A and 1B respectively, including the mean and standard

**Table 1. a. Descriptive statistics of the creativity and its potential predictors including mean, standard deviation (SD), skewness, kurtosis, standard error (SE), and variance inflation factor (VIF). b. Descriptive statistics of the personality trait variables including mean, standard deviation (SD), skewness, kurtosis, standard error (SE), and variance inflation factor (VIF).**

| Variable | N | Mean | SD | Median | Min | Max | Skewness | Kurtosis | SE | VIF |
|---|---|---|---|---|---|---|---|---|---|---|
| Clarity | 3456 | 4.82 | 1.58 | 5 | 1 | 7 | -0.46 | -0.57 | 0.03 | 1.58 |
| Aesthetic Appeal | 3456 | 4.8 | 1.44 | 5 | 1 | 7 | -0.48 | -0.23 | 0.02 | 2.13 |
| Felt Valence | 3456 | 4.5 | 1.62 | 5 | 1 | 7 | -0.41 | -0.48 | 0.03 | 2.59 |
| Felt Arousal | 3456 | 3.86 | 1.73 | 4 | 1 | 7 | -0.14 | -0.92 | 0.03 | 2 |
| Surprise | 3456 | 3.78 | 1.68 | 4 | 1 | 7 | -0.17 | -0.92 | 0.03 | 1.63 |
| Creativity | 3456 | 4.91 | 1.38 | 5 | 1 | 7 | -0.53 | 0.05 | 0.02 | - |
| **Personality Traits** | **N** | **Mean** | **SD** | **Median** | **Min** | **Max** | **Skewness** | **Kurtosis** | **SE** | |
| Openness | 96 | 5.02 | 0.74 | 4.9 | 3 | 6.4 | 0.12 | -0.82 | 0.01 | |
| Intellect | 96 | 4.7 | 0.9 | 4.7 | 2.7 | 6.4 | -0.04 | -0.59 | 0.02 | |
| Awe-proneness | 96 | 5.11 | 1.14 | 5.17 | 1.83 | 7 | -0.48 | -0.04 | 0.02 | |
| Curiosity | 96 | 5.58 | 0.86 | 5.6 | 3.5 | 7 | -0.19 | -0.67 | 0.01 | |

Note: The VIF for a variable is defined for a set of predictor variables by $1/[1-R^2]$ where $R^2$ is the coefficient of determination for the model predicting the variable from all the other predictor variables.

**Table 2. Bivariate correlation coefficients for creativity, its predictors, and the personality measures of the readers.**

| Variable | M | SD | 1 | 2 | 3 | 4 | 5 | 6 | 7 | 8 | 9 |
|---|---|---|---|---|---|---|---|---|---|---|---|
| 1. Clarity | 4.82 | 0.66 | | | | | | | | | |
| 2. Aesthetic appeal | 4.8 | 0.69 | 0.68** | | | | | | | | |
| 3. Felt valence | 4.5 | 0.79 | 0.44** | 0.76** | | | | | | | |
| 4. Felt arousal | 3.86 | 1.19 | 0.25* | 0.47** | 0.64** | | | | | | |
| 5. Surprise | 3.78 | 1.12 | 0.31** | 0.48** | 0.70** | 0.71** | | | | | |
| 6. Creativity | 4.91 | 0.76 | 0.52** | 0.81** | 0.69** | 0.44** | 0.57** | | | | |
| 7. Openness | 5.02 | 0.74 | 0.22* | 0.26** | 0.08 | 0.03 | -0.15 | 0.31** | | | |
| 8. Intellect | 4.7 | 0.9 | 0.27** | 0.35** | 0.1 | 0.05 | -0.03 | 0.31** | 0.43** | | |
| 9. Awe-proneness | 5.11 | 1.15 | 0.25* | 0.31** | 0.29** | 0.13 | 0.13 | 0.36** | 0.47** | 0.36** | |
| 10. Curiosity | 5.58 | 0.87 | 0.30** | 0.35** | 0.27** | 0.11 | 0.12 | 0.41** | 0.33** | 0.47** | 0.57** |

*Note*. *M* and *SD* are used to represent mean and standard deviation, respectively. * indicates $p < .05$. ** indicates $p < .01$. The means and s.d. are over N = 96 but the ratings being averaged for variables 1–6 are first each averaged over the 36 poems before being averaged over the participants.

deviation (SD) for each variable. Table 1A includes five potential predictors, i.e., clarity, aesthetic appeal, felt valence, felt arousal, and surprise, and the outcome variable i.e., creativity. Table 1B includes four chosen personality traits, i.e., openness, intellect, awe-proneness, and epistemic curiosity. The distributions of variables are marginally left-skewed (excepting openness with skewness of 0.12), with low kurtosis values. Variance Inflation Factor (VIF<3) confirms the absence of multicollinearity among the predictor variables [127]. Variance inflation factor (VIF) is a measure of multicollinearity in a multiple regression model indicating whether there is a strong correlation between multiple independent variables in the regression model. The VIF for a variable is defined for a set of predictor variables by $1/[1-R^2]$ where $R^2$ represents the coefficient of determination for the model predicting the variable from all the other predictor variables. If the largest VIF >10 then there is a cause for concern [128,129]; see also [130]. Of note, throughout the article, epistemic curiosity is referred to as curiosity for the sake of clarity and ease of comprehension.

Table 2 shows the bivariate correlations between the poem related predictor variables, personality traits, and creativity. Creativity was positively and significantly (all $p < .01$) correlated with five predictor variables: clarity ($r = 0.52$), aesthetic appeal ($r = 0.81$), felt valence ($r = 0.69$), arousal ($r = 0.44$), surprise ($r = 0.57$). Creativity was also significantly correlated (all $p < .01$) with four personality traits: openness ($r = 0.31$), intellect ($r = 0.31$), awe-proneness ($r = 0.36$), and curiosity ($r = 0.41$). Openness showed no significant correlation with felt valence ($r = 0.08$, $p = 0.46$), arousal ($r = 0.03$, $p = 0.79$), and surprise ($r = -0.15$, $p = 0.15$). Intellect showed no significant correlation with felt valence ($r = 0.01$, $p = 0.34$), and arousal ($r = 0.05$, $p = 0.66$), and surprise ($r = -0.03$, $p = 0.15$). Felt valence was significantly correlated with both awe-proneness ($r = 0.29$, $p = 0.27$) and curiosity ($r = 0.27$, $p = 0.27$). Within personality measures, all were significantly correlated with each other, and the strongest correlation was observed between curiosity and awe-proneness ($r = 0.57$, $p < .01$).

## Parsimonious model formation

We used the forward selection method to determine the order of inclusion of the predictors in the model. The predictor variables were added based on their correlation with the outcome variable, i.e., creativity. The variable with the highest correlation was included first in the null model, followed by the other variables in the descending order of their correlations with creativity, as shown in Table 2. Consequently, the predictor variables were entered into the model

**Table 3. Model comparison to identify the best model fit comprising aesthetic appeal, felt valence, and surprise.**

| Information Criteria | Null Model | Model 1 | Model 2 | Model 3 | Model 4 | Model 5 |
|---|---|---|---|---|---|---|
| AIC | 11160.26 | 9586.52 | 9325.25 | 9032.73 | 9016.77 | 9018.37 |
| BIC | 11178.7 | 9611.12 | 9356 | 9069.62 | 9059.81 | 9067.55 |
| R^2 | 0 | 0.26 | 0.29 | 0.33 | 0.33 | 0.33 |
| Δχ^2 | | 1575.73*** | 263.28*** | 294.51*** | 17.96*** | 0.4 |

Note: Aesthetic appeal, felt valence, surprise, arousal and clarity are included sequentially to Model 1 to Model 5; all models are compared hierarchically, i.e., Model 1 is compared to Null Model, Model 2 is compared to Model 1 and so on; AIC = Akaike Information Criterion; BIC = Bayesian Information Criterion; R^2 = proportion of variation explained by fixed effects [133]; Δχ^2 = Likelihood ratio test statistic for comparison of models. Significance codes: '***' 0.001 '**' 0.01 '*' 0.05.

in the following order: aesthetic appeal, felt valence, surprise, arousal, and clarity. To compare five linear mixed models, we utilized various criteria, including the Akaike information criterion (*AIC*) [131], the Schwarz Bayesian information criterion (*BIC*) [132], the proportion of variance explained by fixed effects ($R^2$) and the Likelihood ratio test statistic ($Δχ^2$). The model comparison results are presented in Table 3. The model (Model 3 in Table 3) comprising aesthetic appeal, felt valence, and surprise (Model 3) demonstrated the optimal fit and parsimony as indicated by a significant likelihood ratio test statistic ($Δχ^2 = 294.51$, $p<0.001$) along with a lower Bayesian Information Criterion (*BIC* = 9069.6) compared to the alternative models. Hence, the model incorporating aesthetic appeal, felt valence, and surprise was deemed the most optimal for predicting creativity.

The linear mixed model result for the best-fitting model is presented in Table 4. Aesthetic appeal was found to be the best predictor ($b = 0.34$, $SE = 0.02$, $t = 22.14$, $p<0.001$), indicating a significant positive relationship with creativity. Following that, surprise significantly influenced creativity ($b = 0.23$, $SE = 0.01$, $t = 17.54$, $p<0.001$), showing a positive association with creativity. Felt valence, although demonstrating a relatively weaker but still significant effect on creativity ($b = 0.16$, $SE = 0.01$, $t = 11.56$, $p<0.001$), was also positively associated with creativity. On the other hand, clarity did not significantly predict creativity ($b = -0.01$, $SE = 0.01$, $t = -0.63$, $p = 0.53$) and was eliminated from subsequent analysis. Furthermore, while arousal exhibited positive association with creativity ($b = 0.07$, $SE = 0.02$, $t = 4.28$, $p < .001$), it did not significantly contribute to improving the model fit ($Δχ^2 = 17.962$, $R^2 = 0.33$). Therefore, arousal was not considered to be the part of our parsimonious model. It is noteworthy that a backward elimination approach supported the validity of this model. In this alternative method, the least correlated variable was systematically removed from the full model. This approach also confirmed that the model incorporating aesthetic appeal, felt valence, and surprise provided the best fit. Therefore, aesthetic appeal, surprise, and felt valence were identified as parsimonious predictors of poetic creativity. Next, we analysed the interaction of the four personality traits with these three predictors.

## Moderating role of the personality traits

We explored the interaction of each of the four personality traits–openness, intellect, awe-proneness, and curiosity–with the three significant predictors of poetic creativity–aesthetic appeal, surprise, and felt valence. Table 5 displays the main effects of the moderators and their interactions with the predictors in the models involving four personality traits.

Openness exhibited significant moderation effect on aesthetic appeal ($b = 0.10$, $SE = 0.02$, $t = 4.83$, $p < .001$), felt valence ($b = -0.06$, SE = 0.02, $t = -3.27$, $p < .001$), and surprise ($b = -0.08$, $SE = 0.02$, $t = -4.76$, $p < .001$) (Fig 1). A significant moderation of intellect was observed

**Table 4. The linear mixed model results for the best-fitting model, comprised of aesthetic appeal, surprise, and felt valence as the predictors of creativity judgment.**

| Fixed Effects | | | | | |
|---|---|---|---|---|---|
| | **Estimate** | **SE** | **df** | **t-value** | **p-value** |
| Predictors | | | | | |
| (Intercept) | 4.91 | 0.08 | 96 | 63.33 | <0.001 |
| Aesthetic appeal | 0.34 | 0.02 | 3360 | 22.14 | <0.001 |
| Felt valence | 0.16 | 0.01 | 3360 | 11.56 | <0.001 |
| Surprise | 0.23 | 0.01 | 3360 | 17.54 | <0.001 |
| Random Effects | | | | | |
| Groups | Variance | SD | | | |
| Participants (Intercept) | 0.56 | 0.75 | | | |
| Residual | 0.73 | 0.85 | | | |
| ICC | 0.43 | | | | |
| N(Participants) | 96 | | | | |
| Observations | 3456 | | | | |
| Marginal R^2 | 0.33 | | | | |
| Conditional R^2 | 0.62 | | | | |

Note. ICC = Intraclass correlation coefficient.
MODEL INFO:
*Observations*: 3456.
*Dependent Variable*: Creativity.
*Type*: Mixed effects linear regression.
MODEL FIT:
*AIC* = 9032.7, *BIC* = 9069.6.
Pseudo-$R^2$ *(fixed effects)* = 0.33.
Pseudo-$R^2$ (total) = 0.62.

on aesthetic appeal ($b = 0.08$, $SE = 0.02$, $t = 4.51$, $p < .001$) with valence ($b = -0.01$, $SE = 0.02$, $t = -0.74$, $p = 0.46$) and surprise ($b = -0.02$, $SE = 0.02$, $t = -1.00$, $p = 0.32$) being unmoderated (Fig 2).

Awe-proneness was found to be a significant moderator on the relationship between creativity and aesthetic appeal ($b = 0.03$, $SE = 0.01$, $t = 2.67$, $p = 0.01$), and surprise ($b = -0.03$, $SE = 0.01$, t $= -2.48$, $p = 0.01$), whereas no significant moderation with valence was observed ($b = -0.00$, $SE = 0.01$, $t = -0.30$, $p = 0.76$) (Fig 3). Finally, curiosity was found to significantly moderate aesthetic appeal ($b = 0.04$, $SE = 0.02$, $t = 2.46$, $p = 0.01$), and surprise ($b = -0.05$, $SE = 0.01$, $t = -3.72$, $p<0.001$), leaving felt valence unmoderated ($b = 0.01$, $SE = 0.02$, $t = 0.84$, $p = 0.40$) (Fig 4).

Consequently, all four personality traits exhibited significant moderation effects on both aesthetic appeal and surprise. However, distinct moderation patterns were observed in these two predictors. The linear positive impact of aesthetic appeal on creativity was strengthened to a greater extent for higher values of the moderators. In contrast, the positive effect of surprise on creativity was attenuated for the higher moderator values The simple slopes analyses results are depicted in Table 6.

Arousal was not included in our parsimonious model as a potential predictor of creativity judgment of poetry. Nevertheless, we recognized the possibility that a predictor might not demonstrate main effect but could still show significant interaction when combined with another factor. Therefore, we examined the interaction effects on arousal. Results are as follows: openness interaction: ($b = -0.01$, $SE = 0.02$, $t = -0.46$, $p = 0.64$); intellect interaction:

**Table 5. Moderation results: Main effects and interactions between personality traits and predictors.**

| Model | Estimate | SE | t | p | Fit [R^2] |
|---|---|---|---|---|---|
| **Openness Model** | | | | | |
| Intercept | 3.32 | 0.5 | 6.58 | <0.001 | |
| Openness | 0.32 | 0.1 | 3.18 | <0.001 | |
| Aesthetic Appeal | -0.15 | 0.1 | -1.46 | 0.14 | |
| Felt Valence | 0.47 | 0.1 | 4.89 | <0.001 | |
| Surprise | 0.66 | 0.09 | 7.2 | <0.001 | |
| Openness*Aesthetic Appeal | 0.1 | 0.02 | 4.83 | <0.001 | |
| Openness*Felt Valence | -0.06 | 0.02 | -3.27 | <0.001 | |
| Openness*Surprise | -0.08 | 0.02 | -4.76 | <0.001 | 0.36** |
| **Intellect Model** | | | | | |
| Intercept | 3.67 | 0.39 | 9.35 | <0.001 | |
| Intellect | 0.26 | 0.08 | 3.21 | <0.001 | |
| Aesthetic Appeal | -0.03 | 0.08 | -0.35 | 0.72 | |
| Felt Valence | 0.22 | 0.08 | 2.8 | 0.01 | |
| Surprise | 0.31 | 0.08 | 3.92 | <0.001 | |
| Intellect*Aesthetic Appeal | 0.08 | 0.02 | 4.51 | <0.001 | |
| Intellect*Felt Valence | -0.01 | 0.02 | -0.74 | 0.46 | |
| Intellect*Surprise | -0.02 | 0.02 | -1 | 0.32 | 0.36** |
| **Awe-proneness Model** | | | | | |
| Intercept | 3.69 | 0.33 | 11.1 | <0.001 | |
| Awe-proneness Model | 0.24 | 0.06 | 3.77 | <0.001 | |
| Aesthetic Appeal | 0.17 | 0.06 | 2.71 | 0.01 | |
| Felt Valence | 0.18 | 0.06 | 2.96 | <0.001 | |
| Surprise | 0.37 | 0.06 | 6.32 | <0.001 | |
| Awe-proneness*Aesthetic Appeal | 0.03 | 0.01 | 2.67 | 0.01 | |
| Awe-proneness*Felt Valence | 0 | 0.01 | -0.3 | 0.76 | |
| Awe-proneness*Surprise | -0.03 | 0.01 | -2.48 | 0.01 | 0.37** |
| **Curiosity Model** | | | | | |
| Intercept | 2.9 | 0.46 | 6.29 | <0.001 | |
| Curiosity | 0.36 | 0.08 | 4.39 | <0.001 | |
| Aesthetic Appeal | 0.06 | 0.1 | 0.63 | 0.53 | |
| Felt Valence | 0.03 | 0.09 | 0.37 | 0.71 | |
| Surprise | 0.5 | 0.08 | 5.99 | <0.001 | |
| Curiosity*Aesthetic Appeal | 0.05 | 0.02 | 2.96 | <0.001 | |
| Curiosity*Felt Valence | 0.02 | 0.02 | 1.43 | 0.15 | |
| Curiosity*Surprise | -0.05 | 0.01 | -3.27 | <0.001 | 0.38** |

($b = 0.03$, $SE = 0.02$, $t = 1.86$, $p = 0.06$); awe-proneness interaction: ($b = -0.01$, $SE = 0.01$, $t = -0.99$, $p = 0.32$); curiosity interaction: ($b = 0.00$, $SE = 0.02$, $t = 0.17$, $p = 0.86$). The findings indicated that influence of arousal on creativity remained unaltered by any of the four moderators.

## Discussion

The present study explored how four personality traits–openness, intellect, awe-proneness, and curiosity–moderate the assessment of creativity in English language poems. We initially identified three key predictors–aesthetic appeal, felt valence, and surprise–from a pool of five potential factors influencing the judgment of poem creativity. We then investigated the interaction between these predictors and participants' personality traits. We found that individuals

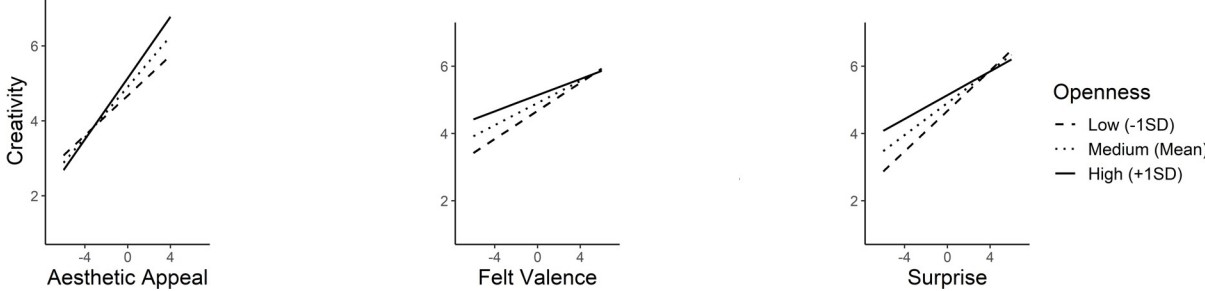

**Fig 1. Simple slopes illustrating significant interactions between openness as the moderator and aesthetic appeal, felt valence, and surprise as the predictors.**

with higher levels of openness, intellect, curiosity, and awe-proneness prioritized aesthetic appeal when assessing the creativity of poems. Notably, only the openness trait showed a moderating effect on felt valence, while the other traits did not demonstrate significant effects.

We identified distinct moderation effects of openness and intellect on the assessment of poetic creativity. Individuals with higher levels of both traits demonstrated a stronger emphasis on a poem's aesthetic appeal when evaluating its creativity, compared to those with lower levels of openness and intellect. Despite being separate traits [92], openness and intellect exhibited a

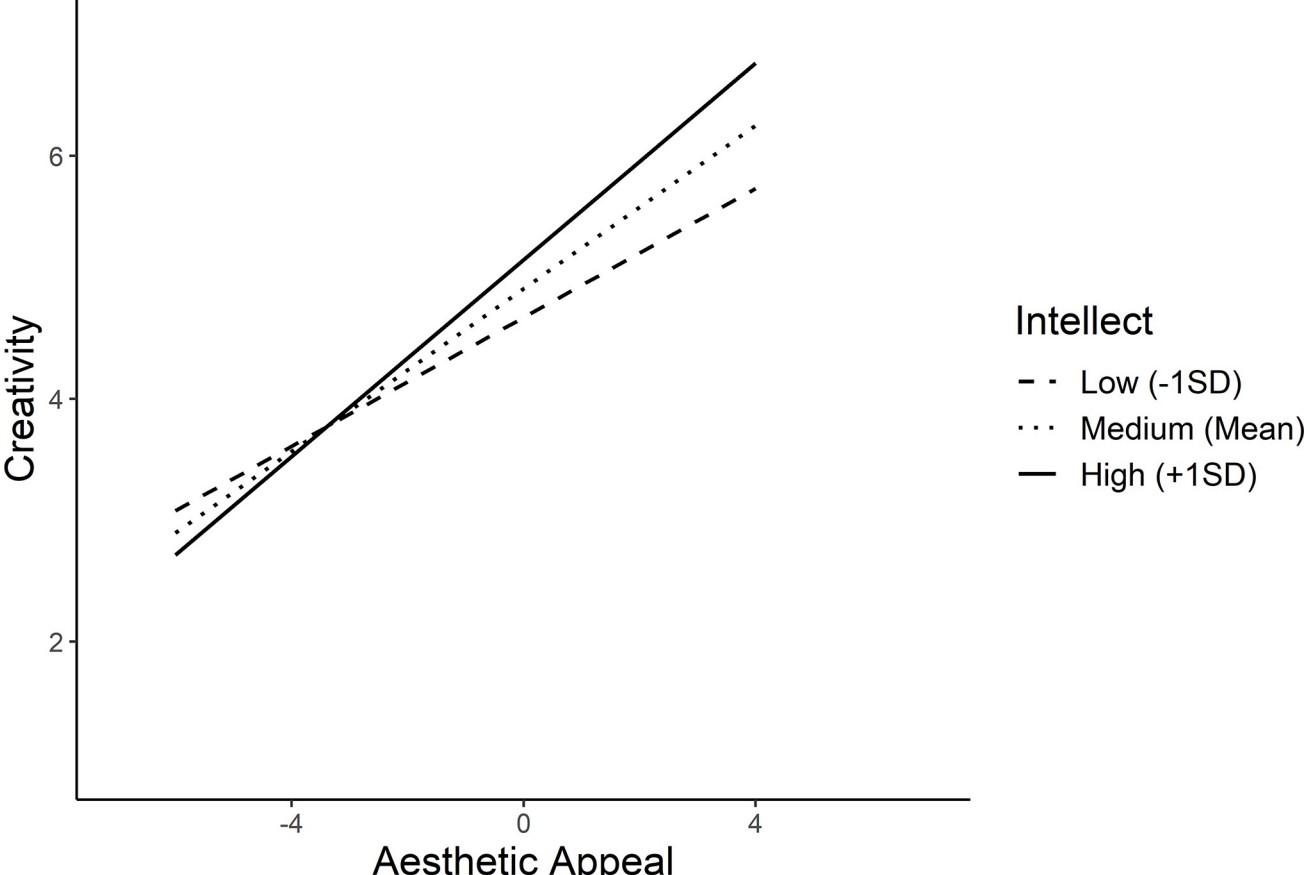

**Fig 2. Simple slopes illustrating significant interaction between intellect as the moderator and aesthetic appeal as the predictor.**

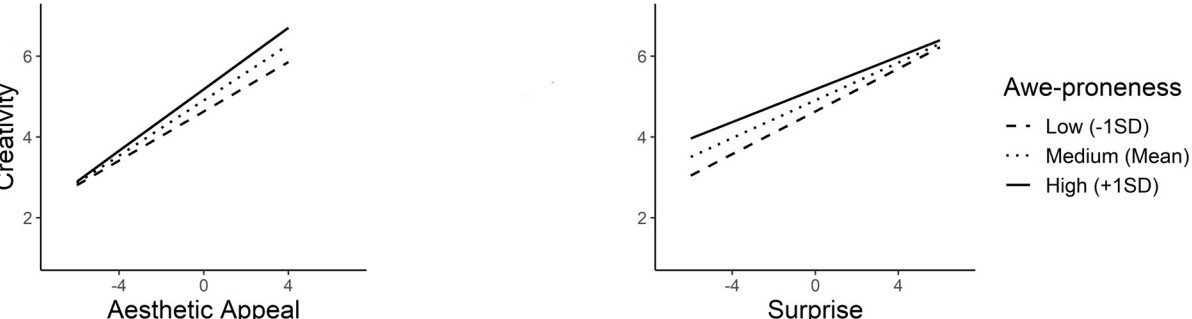

**Fig 3. Simple slopes illustrating interactions between awe-proneness as the moderator and aesthetic appeal and surprise as the predictors.**

shared tendency in appreciating a poem's aesthetic appeal. As aesthetic experience is both style-related and art-specific, involving cognitive and affective processing [134], individuals with higher levels of openness and intellect may have engaged more deeply with both the cognitive and affective aspects during the evaluation process. We postulate that this heightened engagement led them to assign greater significance to the aesthetic appeal of poems in their creativity assessments. Consistent with prior research [90], our study revealed a distinct connection between openness, intellect, and aesthetic appeal. Both openness and intellect seem to reflect a general inclination towards aesthetic experiences—whether it involves processing sensory and aesthetic information (linked to openness) or abstract and complex semantic information (linked to intellect) [85]. Open individuals, i.e., who were assumed to be more unconventional, imaginative, and creative [29,134] exhibited a more pronounced preference for aesthetic appeal in their evaluation of poetic creativity than those with lower levels.

Interestingly, individuals with lower levels of openness appeared to be more influenced by felt valence in their evaluations of poems' creativity compared to those with higher levels of openness. This suggests that readers with higher openness did not weigh their emotional experience during poem reading as heavily as their less open counterparts while judging a poem's creativity. Processing of any artwork, including literature, includes a component called "aesthetic emotion"[134–137]. Aesthetic emotions are the discrete emotions that always include an aesthetic evaluation/appreciation and are further associated with subjectively felt pleasure or displeasure, i.e., felt valence, during any emotional episode [137]. Our study indicates that individuals with higher levels of openness may be less influenced by aesthetic emotions compared to those with lower levels of openness while assessing creativity of poems. On the flip side, higher open individuals seem to be more positively impacted by the overall aesthetic

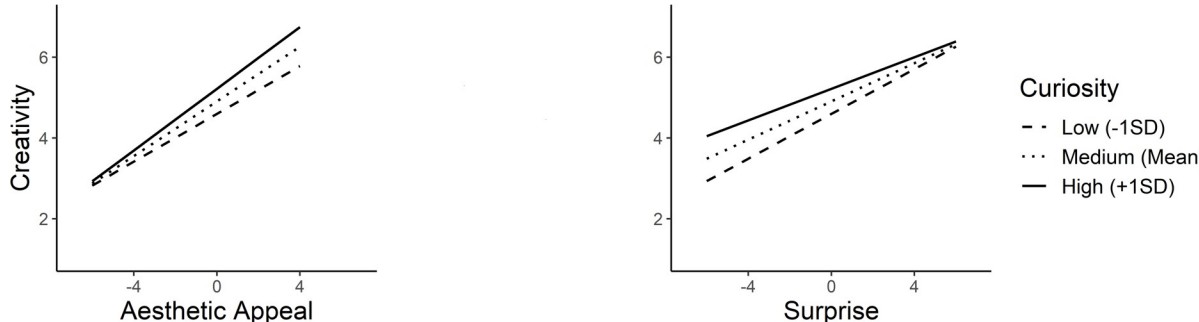

**Fig 4. Simple slopes illustrating interactions between curiosity as the moderator and aesthetic appeal and surprise as the predictors.**

**Table 6. Results of simple slopes analyses for the high and low levels of the moderators and differences in slopes.**

| Predictor | Moderator | High [+1 SD] | | | | Low [-1SD] | | | | Contrast [High-Low] | | | |
|---|---|---|---|---|---|---|---|---|---|---|---|---|---|
| | | Estimate | SE | t-value | p-value | Estimate | SE | t-value | p-value | Estimate | SE | t.ratio | p-value |
| | Openness | | | | | | | | | | | | |
| Aesthetic Appeal | | 0.42 | 0.02 | 20.37 | <0.001 | 0.27 | 0.02 | 12.54 | <0.001 | 0.15 | 0.03 | 5.15 | < .0001 |
| Felt Valence | | 0.12 | 0.02 | 6.62 | <0.001 | 0.2 | 0.02 | 9.72 | <0.001 | -0.08 | 0.03 | -3.08 | 0.0021 |
| Surprise | | 0.17 | 0.02 | 10.61 | <0.001 | 0.3 | 0.02 | 15.68 | <0.001 | -0.13 | 0.03 | -5.25 | < .0001 |
| | Intellect | | | | | | | | | | | | |
| Aesthetic Appeal | | 0.41 | 0.02 | 20.07 | <0.001 | 0.27 | 0.02 | 12.15 | <0.001 | 0.14 | 0.03 | 4.7 | < .0001 |
| | Awe-proneness | | | | | | | | | | | | |
| Aesthetic Appeal | | 0.39 | 0.02 | 18.24 | <0.001 | 0.31 | 0.02 | 15.9 | <0.001 | 0.07 | 0.03 | 2.73 | 0.0063 |
| Surprise | | 0.19 | 0.02 | 11.56 | <0.001 | 0.27 | 0.02 | 14.34 | <0.001 | -0.07 | 0.02 | -3 | 0.0027 |
| | Curiosity | | | | | | | | | | | | |
| Aesthetic Appeal | | 0.39 | 0.02 | 18.78 | <0.001 | 0.31 | 0.02 | 14.59 | <0.001 | 0.08 | 0.03 | 2.95 | 0.0032 |
| Surprise | | 0.19 | 0.02 | 11.52 | <0.001 | 0.28 | 0.02 | 14.69 | <0.001 | -0.09 | 0.02 | -3.63 | 0.0003 |

appeal of poems compared to those with lower levels of openness. This notion aligns with the understanding that aesthetic appeal appreciation and evaluation of artwork, beyond aesthetic emotions, involves processing of other inherent features of art, such as styles, experience of pleasure of generalization [134,138,139], and knowledge [140–142]. Notably, our study demonstrates that levels of intellect have no influence on the positive impact of felt valence on the assessment of creativity of poems.

Individuals with lower levels of openness were found to be more influenced by surprise in their creativity ratings of poems than their higher counterparts. Surprise is often recognized as an interruption mechanism and a short-lived emotion with an unclear positive or negative valence [79]. The statistically significant difference of the simple slopes for high and low open individuals indicates that, more open individuals, who are more motivated to learn, might be less influenced by the surprise in the contents of the poems compared to their lower counterpart while judging poetic creativity. The transient and ambiguous nature of surprise might disrupt their affective states, leading to a reduced impact of surprise on their creativity judgment. In contrast, less open individuals perceived surprise as a more significant factor in their evaluation of poetic creativity than their higher counterparts, contradicting our initial prediction. It is noteworthy that the interaction does not indicate that high openness readers were less surprised by the poems compared to low openness readers. Rather it suggests that their judgments of a poem's creativity were less influenced by the surprise element of the poem compared to those with lower openness. Furthermore, our focus was not on whether individuals with higher openness rated surprise more highly on average than those with lower openness. Instead, we focused on the differential level of surprise ratings for high and low openness. Our objective was to investigate whether there was a difference in how surprise was prioritized as a predictor of creativity judgment between the two levels of openness.

It is worth mentioning that to reach a consensus on how best to define the creativity phenomenon, the 3-criterion definition of creativity [12] is proposed which is based on the three criteria used by the United States Patent Office to evaluate applications for patent protection. This modified definition uses the criteria of novelty or originality, utility or usefulness, and surprise to judge creativity of a product or idea. Our finding indicates that the traditional 3-criterion definition of creativity within the context of poetry may align better with readers who possess lower levels of openness. Thus, our study supports the notion that openness/intellect is an aesthetically sensitive personality domain [90] and consistently serves as a predictor of both

artistic creativity and aesthetic appreciation [23,49,143] across a diverse range of the arts [44,87,96]. Further, this study reveals that, individuals with higher openness and intellect place particular emphasis on the positive impact of aesthetic appeal of poems when evaluating their creativity. However, our findings indicate distinct differences in the moderation effects of openness and intellect when assessing felt valence and surprise in poems during creativity evaluation, emphasizing the nuanced distinction between openness and intellect [92].

Awe-proneness, in our study, demonstrated significant interactions with aesthetic appeal and surprise, but not with felt valence. Awe, a specific emotional response often triggered by beauty, is considered a key member of the self-transcendent emotions [144]. Our findings support the model of apreciation of beauty and excellence [145], which suggests that the ability to perceive and appreciate beauty involves the experience of self-transcendent emotion like awe [144]. Specifically, individuals with higher levels of awe-proneness placed greater emphasis on the aesthetic appeal of a poem when evaluating its creative potential, aligning with the principles of this model. This suggests that readers predisposed to feeling awe might be more sensitive to the artistic and moral beauty of the poems [146], thereby linking dispositional awe to creativity judgment and appreciation for beauty [145,147]. Interestingly, we observed that individuals with lower levels of awe-proneness were more influenced by surprise in their judgments of creativity. Previous research suggests that awe experiences do not require intensive effortful, controlled processing [148], and further, dispositional awe is inversely correlated with the need for cognitive closure [103]. Therefore, our results indicate that in the evaluation of poetic creativity, individuals with higher awe-proneness would prioritize aesthetic appeal while adopting a more passive and receptive stance towards unexpected elements in poetry [149].

Curiosity exhibited significant moderating effects for aesthetic appeal and surprise, mirroring the interaction patterns of awe-proneness. Individuals with heightened curiosity, driven by a desire for new knowledge and experiences [150], demonstrated a more pronounced influence of the aesthetic appeal of a poem on its creativity. This reinforces the idea that curiosity is instrumental in facilitating aesthetic experiences and in the pursuit of understanding complex, abstract, and intellectually challenging stimuli [151]. Additionally, our findings align with previous research indicating that individuals with high trait curiosity tend to find complex poems more comprehensible and engaging [152]. The tendency of highly curious readers to explore unfamiliar aspects of poems may have enhanced their appreciation of aesthetic appeal, contributing to their judgment of creativity. On the contrary, surprise had a stronger impact on creativity judgment among individuals with lower levels of curiosity, contradicting our initial prediction. We anticipated that the positive effect of surprise on creativity scores would be more prominent in those with higher levels of epistemic curiosity. Although literature suggests that surprise can stimulate curiosity [34,35,153], we propose that the way surprise appeared in the poems did not engage the knowledge-seeking behaviour of individuals with higher levels of epistemic curiosity. Rather than facilitating creativity judgment, the unexpected elements in the poems may have been perceived as disruptions, hindering the exploratory and inquisitive mindset of individuals.

The similar interaction patterns between openness and curiosity highlight the well-established link between openness and curiosity [113,154,155]. This indicates that individuals with high openness are more motivated to learn, inclined to explore, and interested in acquiring information. These tendencies might enhance their semantic knowledge [156], and subsequently, their aesthetical experiences [151], and the judgment of poetic creativity. Moreover, similarity in interaction patterns of awe-proneness and curiosity in our results suggest that awe-prone individuals are more curious and that awe itself can stimulate curiosity, which are in line with previous research [157,158]. This further indicates that higher levels of awe-

proneness and curiosity might amplify the perceived ability to comprehend complex stimuli like poetry [152].

It is important to note that this study did not aim to determine whether individuals with higher personality traits tended to rate predictors of creativity more or less favorably on average compared to those with lower traits. Instead, our focus was on examining the differential levels of predictor ratings for readers with high and low traits. We sought to investigate whether there were differences in how these predictors were prioritized between the two levels of personality traits while predicting the judgment of a poem's creativity.

## Limitations

The current study is subject to several limitations. First, we focused on felt emotions, i.e., the emotions experienced by participants while reading poems, rather than perceived emotions, which reflect the perceived emotional quality of the poems. Perceived and felt emotions are not necessarily identical, as highlighted in various studies on music [76,77,159]. We suggest that this is also likely to be the case for poems. For instance, a poem with a 'sad' theme may not necessarily induce sadness in the reader. Of note, previous research has reported an association between perceived valence and aesthetic appeal of poetry [38]. Therefore, future work could investigate the predictive power of perceived emotions on a poem's creativity and the potential moderating role of traits, e.g., intellect. Second, we focused on trait-level personality characteristics rather than state-level personality features. However, contextualized personality traits are crucial for capturing within-individual variability [160]. Future studies should incorporate state-level individual differences to gain a more comprehensive understanding of poetry evaluation. Third, we did not control for various structural elements of poems such as rhythm, form, and genre. We did not impose restrictions on the poems regarding length, rhythmic patterns, or specific forms or genres, such as sonnets, haiku, limericks, or others. However, exploring the specific effects of genres and forms was not feasible due to the limited number of poems in our study, and therefore, the potential influence of these objective features inherent on the creativity assessment could not be ruled out. Fourth, the representativeness of the selected poems may also be limited, potentially impacting the generalizability of our findings. Fifth, concerning the diversity measures of the stimuli, it is important to acknowledge that given the small word count of some of our poems and the implied limited vocabulary, the Type-Token Ratio (TTR) method might not yield reliable results due to constrained variability in word usage within short texts [161,162]. Finally, we assessed the variables using single item measures, a common practice in assessing aesthetics in visual art [163,164], poetry [38,39,116,165,166], and music [167]. However, we also recognize the potential variability in individual interpretation of the questions remains unexplored. Employing multiple items for variable assessment could have offered psychometric advantages, particularly in enhancing reliability and validity [168].

## Conclusions

In summary, our study investigated how specific personality traits, namely openness, intellect, awe-proneness, and epistemic curiosity, influence the evaluation of creativity of English language poetry. We focused on how these traits moderate the impact of three predictors—aesthetic appeal, felt valence, and surprise—in forming a parsimonious model for evaluating poetic creativity. Among the four traits, openness exerted the most significant moderating effect on all three predictors, and among the predictors, aesthetic appeal was significantly moderated by all personality traits in assessing the creativity of poems. These results altogether demonstrate how specific personality traits moderate the underlying model of creativity

judgment of English poems, thereby explaining the variability in individual preferences and evaluations.

## Supporting information

**S1 Table.**
(XLSX)

## Author Contributions

**Conceptualization:** Soma Chaudhuri, Joydeep Bhattacharya.

**Data curation:** Soma Chaudhuri.

**Formal analysis:** Soma Chaudhuri.

**Methodology:** Soma Chaudhuri, Alan Pickering, Joydeep Bhattacharya.

**Supervision:** Joydeep Bhattacharya.

**Writing – original draft:** Soma Chaudhuri.

**Writing – review & editing:** Alan Pickering, Maura Dooley, Joydeep Bhattacharya.

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
