## [Decision Letter · Decision Letter 0]

6 Dec 2023

PONE-D-23-18669Beyond the words: Exploring individual differences in the evaluation of poetic creativityPLOS ONE

Dear Dr. CHAUDHURI,

Thank you for submitting your manuscript to PLOS ONE. After careful consideration, we feel that it has merit but does not fully meet PLOS ONE’s publication criteria as it currently stands. Therefore, we invite you to submit a revised version of the manuscript that addresses the points raised during the review process.

We look forward to receiving your revised manuscript.

Kind regards,

Michael Flor

Academic Editor

PLOS ONE

Here are editor's comments for the manuscript.

Some comments are very light, others require considerable changes.

Also, please read the reviewer comments, they have additional aspects for consideration.

1.

Lines 28-29:

“...of a broad range of English language poems ... rated each poem...”

You have not stated yet how many poems were used. Instead of 'broad range', state the number.

2.

Line 106:

“meter (36, 40), , metaphors (35), (41,42,43), and”

either there is bad formatting, or something is missing on this line.

3.

Line 117:

“we considered both dimensions of emotion – valence and arousal.”

Both 'valence' and 'arousal' are technical term in psychology and related fields.

Please be kind enough to explain those terms to the general audience (a footnote may suffice).

Moreover, since 'felt valence' was found to be important in your work, explain to readers what it is.

4.

Lines 122-125:

“Surprise is considered as a stronger predictor of creativity than value after...”

The description of 'Surprise' dimension is very dry and technical (unlike the other descriptions),

and it does not explain what is 'Surprise' in this context.

Also what is 'value' in the above sentence? Did you mean valence?

Line 127:

“It is a broad range of traits...”

What is “It” ? Why do you begin with “It”? Please formulate the description clearly.

Lines 130-132:

“Openness/intellect … can be differentiated into two major aspects (64): openness and intellect.”

Really? Isn't it a bit tautological?

Please clarify – is it one predictor or two? If those are two predictors, why are they bundled together?

5.

Lines 152-153:

”Earlier research has shown that openness influences attitudes toward targets that violate established schemas (68).”

This is absolutely unclear, needs some elaboration.

What are 'targets'? Targets of what?

Which established schemas? OF behavior? Of Social norms? Of poems? Of poetic structure?

What is meant by 'violate' here?

6.

Line 168:

“aesthetic chill”

Please explain what is aesthetic chill.

7.

Line 196

Please explain what is “Prolific®”

8.

Section “Materials and methods”

This section starts with 'Participants' section and it s quite awkward.

The 'Participants' section should be moved after the 'Materials' section.

9.

Lines 194-196:

The 'Participants' section begins with a description of minimum sample size.

This section needs to be clarified.

What led you to focus on “80% power for a medium effect (^2 = 0.15) with squared multiple 196 correlation ^2 = 0.13.”

It seems rather arbitrary.

10.

Also, you did not specify where the participants came from, in terms of education, profession, English language proficiency (native/non-native) and age groups. Would it be possible to have some indications about those background variables?

11.

Line 196 “healthy participants”

What does 'healthy' mean here? Maybe specify in a footnote. What criteria were used?

Is this aspect really important?

12.

Line 202:

“We described the main experimental procedures for this research to the participants”

It is not clear what was explained. Maybe add an appendix?

How did you explain 'valence' and 'arousal' to the participants?

13.

Section 'Materials':

The procedure of poem selection is not described well.

Selection sources (lines 224-227) should be described before the lexical metrics of selected poems (lines 213-223).

How did you converge on the genres and themes that are reported in Table 1?

Who provided the labels of genres and themes?

14.

Lines 218-221. Concerning “lexical diversity (LD)”.

Explain to readers what is LD, and why it is relevant for this study.

There are many metrics for LD. Please describe which one you used.

The legend of table 1 says “(LD = number of unique words/ total characters).”

That should be explained in the text.

Also, that is not a common metric of LD. Where did you adopt it from?

“The average (SD) LD score for our stimuli set was 14.30 (1.7).”

Switch “(SD) LD” to “LD (DS)”.

Also explain to readers :is 14.30 high or low? What is the scale and what does the value mean?

15.

Lines 222-224:

Explain to readers what is DSI, and why it is relevant for this study.

What is the scale and where on the scale is your value (0.8) located?

16.

Lines 241-242:

“Participants read a poem for 30 seconds”

Explain why such a time limit was set?

Was the poem still visible during creativity rating?

17.

Regarding the questions/measures on four personality traits: as those were shown to be correlated with creativity (Table 3). Please briefly explain the scales and value ranges for the personality traits measures.

Moreover, please describe the distributions of values for all personality measures in your study. For example, As described in Table 2, it seems the four personality traits are also on 7-point scale. Can you establish that you had sufficient variability on the personality scores? It is not clear whether you participants represent sufficient difference on an Openness scale (and other scales). The SD for Openness is just 0.74 – are all participants approximately equally 'open'?.

Do you have sufficient distribution for levels of intellect? Sufficient in terms of representing the general public?

18.

Variable names for personality traits are different in Table 2 and 3. Please adjust.

19.

Table 5 includes a variable “Expertise” but that variable was not described in the texts. Please describe appropriately.

20.

Lines 431-433:

“These results suggest that individuals with higher levels of

imagination, a greater appreciation for aesthetic experiences, and a strong inclination towards art demonstrated...”

This claim is not quite warranted.

Did you measure imagination? “ appreciation for aesthetic experiences” ?

Did you measure inclination towards art?

It seems you did not measure those, so how can you make any claims about those?

21.

Line 465-466;

“Our study suggests that as the level of ambiguity increased with the rise in the aesthetic appeal of a poem,...”

You did not measure levels of ambiguity. So why do you make such unwarranted claims?

22.

Lines 485-486:

“may have been more fostered by the intrinsic drive of highly curious readers to explore the unknown”

To 'explore the unknown' by reading a short poem in 30 seconds? Are you serious?

23.

Lines 488-489

“the positive influence of surprise on creativity...”

'Positive influence' is a very ambiguous and term. Consider rephrasing.

24.

Lines 494-496:

“Firstly, it suggests that emotions such as

amazement, wonder, information-seeking curiosity, and knowledge hunger, when

experienced at higher levels, have a stronger impact”

You did not measure any of those. Why are you extrapolating?

25.

Please revise the Discussion section with relevant claims and avoid any unwarranted and whimsical claims. Please try to make the discussion more scientific and less poetic.

26.

Regarding the lack of effect of 'arousal'. Could it be that participants did not understand what it is /supposed to be?

27.

About limitations (from line 524) – please set a new section with title “Limitations”.

Journal Requirements:

Reviewers' comments:

Reviewer's Responses to Questions

**Comments to the Author**

1. Is the manuscript technically sound, and do the data support the conclusions?

Reviewer #1: Yes

2. Has the statistical analysis been performed appropriately and rigorously? 

Reviewer #1: Yes

3. Have the authors made all data underlying the findings in their manuscript fully available?

Reviewer #1: Yes

4. Is the manuscript presented in an intelligible fashion and written in standard English?

Reviewer #1: Yes

5. Review Comments to the Author

Reviewer #1: The study investigates the influence of personality traits on the relationship between various measures of aesthetic experience and the evaluation of creativity in English-language poems. The results of the study suggest that the character trait "openness" in particular has a significant influence on the relationship between the various measured dimensions of aesthetic experience and creativity. The authors conclude that the relationship between aesthetic experience and the evaluation of creativity is not exclusively idiosyncratic, but is systematically mediated by certain personality traits.

The study is very interesting and highly relevant to the empirical study of aesthetics in poetry. The study design is appropriate to the research question and the implementation meets scientific standards. In summary, I can endorse the publication of the study.

I have only minor comments concerning the presentation of the theory and the discussion of the results. However, there are a few things that I think should be changed urgently before publication.

1: Used terminology

Most problematic is, in my view, the use of the term “creativity”. In my opinion, creativity is a state of mind or a quality of people (or – more precisely – a quality of humans that allows them to create something new). Weisberg – for example – defines “creativity” as (a) “the capacity to produce [novel] works [with value]” and (b) “the activity of generating such products” (in: Robert W. Weisberg: Creativity, p. 4). But it is NOT a property of works of art. However, in the sentence: "... different individuals may hold contrasting views on the creativity of a particular poem ...", creativity is spoken of as if it were an inherent property of poems, not an ability of the person who wrote the poem. Similar is the use of the word throughout the article.

This is relevant to the article in that it is not entirely clear what exactly is being measured. Particularly, as the concept and its operationalisation are not explained anywhere. The item used in the questionnaire to assess "creativity" again treats it as an inherent quality of the poems. For a better understanding of the study, it would therefore be imperative to discuss the unconventional use of the term, its implied meaning and its possible impact on the participants of the study.

In the literature cited in the manuscript “creativity” is - as far as I know – also seen as a characteristic of people (i.e., the ability of people to be creative). The connection between creativity and basic character traits found in other studies is therefore only indirectly related to this study, if at all. In other studies, various characteristics of an individual were related to each other (e.g., is extroversion related to creativity?). The present study, on the other hand, investigated which character traits of the viewers had an influence on the evaluation of the poems as ... - well, what exactly?

I think it is therefore urgent to (a) explain how the term "creativity" is used in the article, (b) critically analyse the relationship between the use of the term in the article and in the cited literature, and (c) discuss in detail the potential effects of its unconventional operationalisation on the results of the study.

Note: Despite my reservations, I use the term "creativity" in a similar way as it is used in the manuscript. However, this is simply for lack of alternatives, as I am not quite sure what exactly was measured in the study.

2: Operationalisation of the factors:

The other variables for recording the aesthetic quality of the poems were also - if I understand correctly - assessed by only one item. Here, too, the question arises as to whether the operationalisation of the question is appropriate. Do the study participants understand what is meant when they have to evaluate whether a poem is "clear"? To what extent does the interpretation of the question vary individually? Are there studies that have examined the validity of these items to capture the underlying concept?

3: Selection of poems:

The number of poems used for the study is rather small, 36, and - probably - not representative. Especially since the poems were "carefully selected" (by whom, actually?), it is to be feared that the choice of poems might have influenced the result. I understand that this is a necessary sacrifice to make in empirical studies. Especially since the study design almost certainly excludes an influence of the selection of the poems on the result. But I think the issue should at least be discussed as one of the shortcomings of the study. It would also be helpful if it could be shown what measures were taken to ensure that the selection was not biased.

Minor issues:

1. Line 49: It is not clear to me what you mean by “poetry and its creative evaluation has not been adequately characterized”. First of all, it seems to me that it is not the "creative evaluation" that needs further investigation, but rather the "evaluation of creativity".

It also seems that the authors have not considered a whole field of science that deals with this issue of creativity in poems. The authors could look at studies on foregrounding or the field of stylistics. Another research approach that would be relevant in this context is the work of Menninghaus and colleagues.

2. Line 50 to 54: It is also not entirely clear to me why the subjectivity of aesthetic evaluation should be more pronounced in poetics than in other forms of art. Personal experience always plays a role in the reception of art, regardless of the medium or art form, doesn't it?

Furthermore, it seems to me that the introduction does not clearly separate the concepts of "beauty", "aesthetics" and "creativity". “Beauty” is in the eyes of the beholder. Why then do different observers have different views of the “creativity” with which a poem was written? Are "beauty" and "creativity" the same thing? Or are they causally linked? And if we already know that, what is the point of the study?

3. Line 62-64: In much of the introduction, the authors make assertions without citing any valid references. For example, I do not understand why "creativity" should be a less tangible construct than "preference".

4. Line 79 (and others): Some sentences end without a period or have commas before the period (line 102).

5. Line 457-467: In the discussion, an indirect connection is made between the ambiguity of expression in a poem and aesthetic appeal via the evaluation of "awe-proneness". It is stated that "other studies" have reported a connection between "awe-proneness" and the ability to tolerate ambiguity. But, on the one hand, this seems to concern only one study, and on the other hand, this is an indirectly established connection. I therefore think it is appropriate to formulate the conclusion (which I do not want to doubt) more cautiously (lines 465 and 466).

In order to substantiate the thesis a little, it might be helpful to support it a little with findings from the studies on "foregrounding" (British stylistics or Russian formalism).

6. PLOS authors have the option to publish the peer review history of their article (what does this mean?). If published, this will include your full peer review and any attached files.

Reviewer #1: No

---

## [Author Response · Author response to Decision Letter 0]

17 Jan 2024

Our responses to the Editor’s comments

1. Lines 28-29: “...of a broad range of English language poems ... rated each poem...” You have not stated yet how many poems were used. Instead of 'broad range', state the number

Our response: Thank you for your suggestion. The number of poems has now been mentioned in the manuscript, as follows: “This study investigated how personality traits, specifically openness, intellect, awe-proneness, and epistemic curiosity, shape the implicit associations between subjective experiences and the judgment of creativity of thirty-six English language poems.” 

2. Line 106: “meter (36, 40), , metaphors (35), (41,42,43), and” either there is bad formatting, or something is missing on this line.

Our response: Thank you. Formatting has been revised. 

3. Line 117: “we considered both dimensions of emotion – valence and arousal.” Both 'valence' and 'arousal' are technical term in psychology and related fields. Please be kind enough to explain those terms to the general audience (a footnote may suffice). Moreover, since 'felt valence' was found to be important in your work, explain to readers what it is.

Our response: Thank you for the suggestion. We explained the terms in the revised manuscript as suggested. The lines read as follows: 

“According to the two-dimensional circumplex model of emotion, emotional states are represented along two orthogonal dimensions: valence (pleasure-displeasure: horizontal axis) and arousal (degree of arousal: vertical axis) (Russell, 1980). Poetry evokes strong emotions (Wassiliwizky et al., 2017) and emotional states can influence creativity evaluation (Mastria et al., 2019)…. Of note, perceived and felt emotions may be distinct (Gabrielsson, 2001; Marin & Bhattacharya, 2010). Perceived emotions are the emotions evoked by the stimuli whereas felt emotions are the emotions felt by the perceivers. In our study, we focused on the felt emotions, i.e., the emotions felt by the reader while reading the poem, rather than the perceived emotion, i.e., the emotion recognized in the poem. Therefore, felt valence reveals the extent to which the readers felt positive or negative emotions while reading the poems, whereas felt arousal here reveals how intense it was felt by the readers.” 

4. Lines 122-125: “Surprise is considered as a stronger predictor of creativity than value after...” The description of 'Surprise' dimension is very dry and technical (unlike the other descriptions), and it does not explain what is 'Surprise' in this context. Also what is 'value' in the above sentence? Did you mean valence?

Our response: Thank you very much for the comment. We have now expanded our explanation to clarify the specific meaning of “surprise’ within the context of our study. The section reads as: “Surprise is usually a short-lived emotion elicited by events that deviate from an established schema (Meyer et al., 1997; Meyer et al., 1991; Noordewier & Breugelmans, 2013). Of note, a schema, in this sense, refers to a component of the organism’s knowledge structure, activated by a specific stimulus (Rumelhart, 1984). Surprise is recognized as a key predictor of creativity (Simonton, 2012; Acar et al., 2017) and is also a robust predictor of aesthetic judgment of artwork (Pietras & Ganczarek, 2022). In the context of our study, the term “surprise” specifically denotes the degree to which there is a sudden and unexpected change in the context or theme of the poem.”

Response to your next question on “value”: Despite the lack of an agreed-upon definition, a general consensus over a standard definition of creativity (Runco & Jaeger, 2012; see also Stein, 1953; Barron, 1955) involves two factors: The first factor consists of characteristics such as novelty, originality, infrequency, or unusualness; and the second factor is related to usefulness, value, utility, effectiveness, adaptability, or appropriateness (Acar et al., 2017). However, we excluded the line containing “value” to avoid confusion of readers as we did not explore any potential consequences of this term later in the article. 

Line 127: “It is a broad range of traits...” What is “It”? Why do you begin with “It”? Please formulate the description clearly.

Our response: The line has been amended as: Openness to experience is a broad range of traits, from intellectual abilities to aesthetic and artistic interests (Oleynick et al., 2017;DeYoung et al., 2012; Chamorro-Premuzic et al., 2009) and is most robustly associated with measures of creativity (McCrae, 1996).

Lines 130-132: “Openness/intellect … can be differentiated into two major aspects (64): openness and intellect.” Really? Isn't it a bit tautological? Please clarify – is it one predictor or two? If those are two predictors, why are they bundled together?

Our response: Actually, Costa Jr & McCrae, (1992) first introduced Openness to Experience as the fifth factor of personality traits. Later, DeYoung et al created the Openness/Intellect (O/I) model, which proposed that Openness to Experience has two major facets– the two “related but separable trait dimensions” (DeYoung et al., 2007, p.883): openness and intellect. Based on earlier work on the neurocognitive correlates of Openness to Experience (DeYoung et al., 2005), DeYoung et al., 2009 hypothesized that brain regions associated with working memory would predict the intellect facets but not the openness facets. Showing distinct effects of openness and intellect would provide evidence for the validity of both constructs (see Nusbaum & Silvia, 2011).

Hence, in our study, we are interested to explore the distinct influence of openness and intellect on the evaluation of creativity of poems. 

5. Lines 152-153: “Earlier research has shown that openness influences attitudes toward targets that violate established schemas (68).” This is absolutely unclear, needs some elaboration. What are 'targets'? Targets of what? Which established schemas? OF behavior? Of Social norms? Of poems? Of poetic structure? What is meant by 'violate' here? 

Our response: We appreciate your comment. We revised the lines as: “As surprise is often experienced when individuals encounter unexpected or schema-discrepant events (Meyer et al., 1997; Meyer et al., 1991), we assumed that the assessment of expectancy violations in the judgment of poetic creativity necessitates a substantial degree of cognitive engagement. Consequently, we predicted that both Openness and Intellect would moderate the relationship between surprise and creativity. ”

6. Line 168: “aesthetic chill” Please explain what is aesthetic chill.

Our response: We explained the term. The line now reads: “Aesthetic chills are transient emotional responses (McCrae, 2007) to aesthetical stimuli. They can be described as experiencing a chill or wave of excitement when engaging with poetry or looking at a work of art (Costa & McCrae, 2008).”

7. Line 196: Please explain what is “Prolific®” 

Our response: Explained as suggested: “The experiment was created using Qualtrics®, and the link was disseminated through Prolific®, a platform for participant recruitment.” 

8. Section “Materials and methods” This section starts with 'Participants' section and it’s quite awkward. The 'Participants' section should be moved after the 'Materials' section.

Our response: Thank you. “Participants” section is now shifted after the “Materials” section, as suggested.

9. Lines 194-196: The 'Participants' section begins with a description of minimum sample size. This section needs to be clarified. What led you to focus on “80% power for a medium effect (^2 = 0.15) with squared multiple correlation ^2 = 0.13.” It seems rather arbitrary.

Our response: The section has been clarified as suggested. The section now reads as: 

“By using the G*Power software (v. 3.1.9.4), (Faul et al., 2007) we found that a minimum sample size of 98 was required to detect a medium effect size (f2 = 0.15) in a multiple linear regression, assuming a significance level of 0.05 and a statistical power of 80%. The criteria we used are widely-used conventional figures when estimating sample sizes. Additionally, we used the ‘samplesize_mixed’ function in R, which estimates the required sample size for linear mixed models (two-level-designs), based on power-calculation for standard design and adjusted for design effect for 2-level-designs (https://strengejacke.github.io/sjstats/). For our multilevel model, which considered 96 cluster groups and a small to medium effect size (Cohen’s d = 0.3), with 36 observations per cluster group (level-2-unit), we determined that a total of 965 observations would be necessary. This translates to a minimum of 27 participants (965/36).” 

10. Also, you did not specify where the participants came from, in terms of education, profession, English language proficiency (native/non-native) and age groups. Would it be possible to have some indications about those background variables?

Our response: As per the self-reported demographic questionnaires, all 96 participants (32 males, 63 females, 1 preferred not to say; mean age = 31.94 years, SD = 13.09) were fluent in English and came from a variety of educational backgrounds, with each holding at least a bachelor’s degree in any discipline. Additionally, our sample represented a diverse range of professions and ethnicities. 

We included this information in the “Participants” section. 

11. Line 196: “healthy participants” What does 'healthy' mean here? Maybe specify in a footnote. What criteria were used? Is this aspect really important?

Our response: Thank you for your comment. Considering that the “health” aspect was not a significant focus of our study, we omitted the term “healthy.” 

12. Line 202: “We described the main experimental procedures for this research to the participants” It is not clear what was explained. Maybe add an appendix? How did you explain 'valence' and 'arousal' to the participants?

Our response: The instructions to the participants are now included in the main manuscript. The full paragraph now reads:

“Participants were briefed about the experimental procedure, which involved the assessment of a set of English poems on a 7-point Likert rating scale (1=extremely low; 7=extremely high) across various constructs, including clarity, aesthetic appeal, felt valence, arousal, surprise, and overall creativity. Additionally, participants were instructed to complete demographic and personality-related questions. We assured participants of the full confidentiality of their data, in compliance with the General Data Protection Regulation, and clarified that any published results would be non-identifiable. All participants provided informed consent (online) before data collection. Participants received a cash incentive of £7.50 per hour. The data collection period spanned from 27 January 2022 to 23 June 2022, and the data were accessed for research purposes only after this period.”

How did you explain 'valence' and 'arousal' to the participants?

For felt valence, participants were asked: “How positive (higher Scores) or negative (lower Scores) did you feel when you read the poem?” For arousal, they were asked: “How stimulating (higher Scores) or relaxing (lower Scores) did you feel when you read the poem?”

13. Section 'Materials': The procedure of poem selection is not described well. Selection sources (lines 224-227) should be described before the lexical metrics of selected poems (lines 213-223). 

Our response: As suggested, the sources for poem selection have been described in the “Material” section:

“Initially, we selected 108 poems from various online resources, including Poetry.org (http://www.poetry.org/), the Poetry Foundation (https://www.poetryfoundation.org/), and the Academy of American Poets (https://poets.org/). These poems were subsequently rated for "surprise" on a scale of 1 to 7 by M.D, one of the co-authors who is a Professor of Creative Writing with domain-specific expertise and an award-winning poet. Following this evaluation, we shortlisted 36 as the experimental stimuli: 18 with low surprise ratings (4 or lower) and 18 with high surprise ratings (6 or above). The chosen poems varied in structure, contents, lines, and word count (mean number of lines = 11, SD = 3.2; mean word count = 71.3, SD = 29). To represent a broad spectrum of English poems, we consciously chose not to limit our selection to a particular genre or form, like haiku or sonnets as done in previous studies (Belfi et al., 2018; Hitsuwari & Nomura, 2022; (Papp-Zipernovszky et al., 2021). 

13. How did you converge on the genres and themes that are reported in Table 1? Who provided the labels of genres and themes?

M.D, a co-author of our study who is a Professor of Creative Writing with domain-specific expertise and an award-winning poet, validated the genre and theme-specific information about the stimuli.

14. Lines 218-221. Concerning “lexical diversity (LD)”. Explain to readers what is LD, and why it is relevant for this study. There are many metrics for LD. Please describe which one you used. The legend of table 1 says “(LD = number of unique words/ total characters).” That should be explained in the text. Also, that is not a common metric of LD. Where did you adopt it from? “The average (SD) LD score for our stimuli set was 14.30 (1.7).” Switch “(SD) LD” to “LD (DS)”. Also explain to readers: is 14.30 high or low? What is the scale and what does the value mean?

Our response: Thanks for this valuable comment. While lexical diversity and semantic diversity may not be directly pertinent to our study, we decided to include them to capture the variedness and richness of vocabulary and semantics in our chosen stimuli. Of note, we rectified the lexical diversity scores (computed in R). Table1, originally presenting stimulus details, has been modified to incorporate updated Lexical Diversity and is now designated as Table S1. We intend to include it in the Supplementary section.The paragraph describing the diversity metrics has been modified as follows:

“Lexical diversity (LD) of a text refers to its lexical richness, indicating range and variety of vocabulary deployed in a text (McCarthy & Jarvis, 2007). For computing the LD of the poems, we used the type-token ratio (TTR) method, which calculates the ratio of unique words (types) to the total word count (tokens) (Chotlos, 1944). It ranges from 0 to 1: a higher TTR indicates a greater lexical diversity. The average lexical diversity across the poems is 0.77 with a standard deviation of 0.09, suggesting that, on average, about 77% of the words used in the poems are unique or different. Semantic diversity, on the other hand, refers to the range of contexts (i.e., semantic richness) in which words are used (Johnson et al., 2022). To calculate the semantic diversity, we used divergent semantic integration (DSI) (http://semdis.wlu.psu.edu/), representing the mean semantic distance between all word pairs in a text; DSI varies from 0 to 1, where a higher score indicates that a broader collection of divergent ideas. The average DSI of our stimuli is 0.80 with a standard deviation of 0.03, indicating a high degree of semantic variety (see Table S1 in the Supplementary section for details).”

In the “Limitations” section, we added the limitations of TTR method for short texts as follows: “it is important to acknowledge that given the small word count of some of our poems and the implied limited vocabulary, the Type-Token Ratio (TTR) method might not yield reliable results due to constrained variability in word usage within small texts (Malvern & Richards, 1997; McCarthy & Jarvis, 2010).” 

The table (Table S1 in the Supplementary section) containing the details of the stimuli will be submitted in the Supplementary section and OSF (https://osf.io/9mw7r/?view_only=07137f4871d146c790501f22bc7743d5).

TableS1. The details of the stimuli

15. Lines 222-224: Explain to readers what is DSI, and why it is relevant for this study. What is the scale and where on the scale is your value (0.8) located?

Our response: We explained DSI in the “Materials” section as follows: 

“Semantic diversity, on the other hand, refers to the range of contexts (i.e., semantic richness) in which words are used (Johnson et a

---

## [Decision Letter · Decision Letter 1]

8 Mar 2024

PONE-D-23-18669R1Beyond the words: Exploring individual differences in the evaluation of poetic creativityPLOS ONE

Dear Dr. CHAUDHURI,

Thank you for submitting your manuscript to PLOS ONE. After careful consideration, we feel that it has merit but does not fully meet PLOS ONE’s publication criteria as it currently stands. Therefore, we invite you to submit a revised version of the manuscript that addresses the points raised during the review process.

We look forward to receiving your revised manuscript.

Kind regards,

Michael Flor

Academic Editor

PLOS ONE

Journal Requirements:

Additional Editor Comments (if provided):

The revised manuscript is much better than the previous version.

But there are still some issues.

Please read the comments from both reviewers.

I have a few minor comments, and some major ones.

1.

Lines 207-209:

“we shortlisted 36 poems as the experimental stimuli: 18 with low surprise ratings (4 or lower) and 18 with high surprise ratings (6 or above).

The ratings '4 or lower and 6 or above' – who gave them ? M.D.? What scale was used? Please describe shortly.

2.

Line 273: “onpersonality traits” - space missing

3.

Major issue.

There is a problem with Table 1 "Descriptive statistics".

It shows statistics for 10 variables, all of them measured with 7-point scales .

However the variables are not of the same type.

The first 6 variables are ratings on poems, each on a 7-point scale as stated in the manuscript earlier.

Their count (n) values are 3456 data points, that is36 poems x 96 participants.

However the last four variables are personality measures, their count (n) should be 96, not 3456.

I presume that was a 'typo', but:.....

Please recheck your statistics carefully!!!

Please involve all co-authors in rechecking the manuscript.

Please separate Table 1 into two (similar) tables (or table 1a and 1b), one for ratings on poems, and one for personality traits. Include those labels on table captions.

Also mention in the manuscript explicitly that personality traits were also measured on 7-point scales (it is hidden in the extra materials).

4.

Major issue.

Concerning figures 1-4 -all the figures with slopes. Please explain (also to the readers) what are the scales on X- and Y- axes. The original measurement scales were 7-point scales. On the figures we meet what seems to be re-centered scales, but they are not obvious, and in some cases the ranges are -6 to 4, which is strange. One would expect the ranges to be of size six. Explain your transformations.

5.

Lines463-465:

"The current study investigated the moderating effects of four personality traits –

openness, intellect, awe-proneness, and curiosity - on the evaluation of creativity in English

language poems: openness."

What is 'openness' doing at the end of the sentence?

6.

Lines 495-498:

"The distinct moderating effects of openness on aesthetic appeal and

felt valence in our study suggest that the aesthetic emotions evoked through affective

processing of poetry might differ from the emotional valence experienced during the reading

of poetry."

I think I understand every word in this sentence, but I don't understand what it means.

What is the difference between "affective processing of poetry" and "emotional valence experienced during the reading of poetry" ?

Please try to clarify this sentence, and the whole discussion section,

stating things in very plain language, and taking in mind an audience who are not versed in your jargon.

7

Line 525: "a finding corroborated by previous research (111)."

Please reconsider your semantics. Your study might corroborate previous research, but claiming that previous research corroborates your study is over the top.

8.

Regarding lines 534-557. There can be an alternative 'explanation': individuals with high openness that are more motivated to learn, might know more, and thus be less surprised in general. They might also be less surprised by the contents and forms in the poems. This could be checked with your data: are the average surprise ratings of open individuals lower than surprise rating of not-open individuals?

Do low-openness individuals put more emphasis on surprise, or maybe they are just more surprised? Or both? (the two things are not the same).

9. Major issue.

The discussion section contains too much speculation, some of which could actually be checked with existing data. Perhaps in another paper?

For this paper, please tone down the speculations.

I will restate it in different terms: when reading the discussion, I did not feel awe, just confusion.

One of the reviewers complained on the lack of conceptual clarity. I think it acutely applies to the discussion. Make the discussion clearer and less speculative.

Another aspect of the discussion:

it contains mostly abstract words, many of them nominalizations, and many key nouns are modified by adjectives. This makes reading more difficult. Try writing this section in shorter sentences, it might help. Try using less subordinated clauses.

10.

Lines554-557:

"Our results also suggest that awe-prone individuals are more

curious (120), and that awe itself can stimulate curiosity (121)."

When one says "results suggest" it implies something new; but you have references, so it is old.

Consider rephrasing.

11.

Lines590-591:

"We, here operationalized ‘creativity’ as the poem’s creative potential."

Regarding the issue, raised by a reviewer, on whether creativity is a property of people or of results. It is a difficult issue, but a notion of 'creative potential' is not likely be understood by the audience.

Here is a suggestion, by twisting the 4P approach. Results can often be called 'creative', not by attributing them creativity per se, but just by implicitly denoting them as results of a creative process. Such implicit attribution is pervasive, so we call many things 'creative' without implying that the things can create something.

Reviewers' comments:

Reviewer's Responses to Questions

**Comments to the Author**

1. If the authors have adequately addressed your comments raised in a previous round of review and you feel that this manuscript is now acceptable for publication, you may indicate that here to bypass the “Comments to the Author” section, enter your conflict of interest statement in the “Confidential to Editor” section, and submit your "Accept" recommendation.

Reviewer #1: All comments have been addressed

Reviewer #2: (No Response)

2. Is the manuscript technically sound, and do the data support the conclusions?

Reviewer #1: Yes

Reviewer #2: Partly

3. Has the statistical analysis been performed appropriately and rigorously? 

Reviewer #1: Yes

Reviewer #2: I Don't Know

4. Have the authors made all data underlying the findings in their manuscript fully available?

Reviewer #1: Yes

Reviewer #2: Yes

5. Is the manuscript presented in an intelligible fashion and written in standard English?

Reviewer #1: Yes

Reviewer #2: Yes

6. Review Comments to the Author

Reviewer #1: All my comments have been answered to my satisfaction. Nevertheless, I believe that it would be in the authors' own interest to explain their definition of the term "creativity" in more detail (and to do so at the beginning of the paper). I believe that a large proportion of readers do not know in which tradition the authors use the term. To avoid losing these readers, it would be helpful to explain what the "4P model" is and how to define creativity "through the lens of the product". Honestly, I believe that the majority of readers - like me - will not understand a single word when they read this.

I would also recommend that you invest a little time to correct typos. To give some examples: Line 91: "artwork..", line 273: "onpersonality", line 465: "openess", ...

Reviewer #2: This is generally fluently written and presents the background fairly well. Although the 'gap' this fills is pointed out, it could be strengthened in terms of how poetry might differ from other artforms that may have already garnered similar research in the past.

There's a lot going on in this article, and the justification for why the authors have chosen these particular 4 personality traits, and the specific outcome variables that they have, could have been more strongly justified.

Because there were so many variables being considered in relation to one another, it started to feel a little like a fishing expedition. Although the authors did justify and explain most of their reasoning, it was often hard to follow and since there were so many similarities between the variables, it began to feel a bit unclear to me why there needed to be the level of complexity and number of variables that there were. I was confused by the variables that were measured when selecting the stimuli - expert-rated 'surprise' of the poems (but the participants also rated this themselves?), and lexical diversity/divergent semantic integration measures - was this just to ensure that the stimuli set was diverse? Or were these to be factored into analysis at the end as well somehow (but never were)? I was also confused about the measure of 'felt valence' on a Likert scale from low to high. Valence is about pleasantness vs unpleasantness - is the assumption that high corresponds to pleasant? What about a sad poem that evokes sadness in the participant? Is that low felt valence? The interpretation of this variable seemed to be more about the level of emotion (either pleasant or unpleasant), rather than true valence, so seemed closer actually to the arousal variable, rendering the interpretation of this variable in relation to the others confusing.

Because there are so many variables, my main concern from the start was multicollinearity. Regression is not my area of expertise, so I defer to other colleagues with more knowledge than I do hopefully on this, but it concerns me that almost all of the variables seem to be highly correlated with one another in Table 2 yet the authors suggest that there is no multicollinearity? I'm not familiar really with exactly what the VIF test does, but on a theoretical level, constructs like openness to experience and curiosity are acknowledged as being very closely related by the authors, and both found to be predictive of creativity in previous literature, but they are being treated as separate predictor variables, and I'm not sure how useful that really is then to the overall narrative of this study.

Overall, I think there's just too much going on here without enough conceptual clarity to justify the complexity. I find the overall discussion of design and findings both hard to follow and, at their core, a bit unsurprising really and its novelty and indeed accuracy are not clear to me, so I'm afraid I am not inclined to recommend this for publication.

I have made comments and corrections on the PDF of the article itself along the way, which I hope will be helpful to the authors in future drafts, should this end up published or they rework it and take it elsewhere.

7. PLOS authors have the option to publish the peer review history of their article (what does this mean?). If published, this will include your full peer review and any attached files.

Reviewer #1: No

Reviewer #2: No

---

## [Author Response · Author response to Decision Letter 1]

15 May 2024

Responses to the Editor’s comments

1.Lines 207-209: “we shortlisted 36 poems as the experimental stimuli: 18 with low surprise ratings (4 or lower) and 18 with high surprise ratings (6 or above). 

Editor’s comment: The ratings '4 or lower and 6 or above' – who gave them? M.D.? What scale was used? Please describe shortly.

Our response: Thank you for your valuable comment. The initially selected poems were evaluated for their levels of "surprise" by M.D., a Professor of English and Creative Writing with domain-specific expertise, using a scale of 1 to 7, where 1 indicates “absolutely not surprising” and 7 indicates “absolutely surprising.” This has now been incorporated in the Materials and Methods section. 

2. Line 273: “onpersonality traits” - space missing

Our response: Thank you. This has been corrected. 

3. Major issue. There is a problem with Table 1 "Descriptive statistics". It shows statistics for 10 variables, all of them measured with 7-point scales . However the variables are not of the same type. The first 6 variables are ratings on poems, each on a 7-point scale as stated in the manuscript earlier. Their count (n) values are 3456 data points, that is36 poems x 96 participants. However the last four variables are personality measures, their count (n) should be 96, not 3456. I presume that was a 'typo', but:..... Please recheck your statistics carefully!!! Please involve all co-authors in rechecking the manuscript. Please separate Table 1 into two (similar) tables (or table 1a and 1b), one for ratings on poems, and one for personality traits. Include those labels on table captions. Also mention in the manuscript explicitly that personality traits were also measured on 7-point scales (it is hidden in the extra materials).

Our response: Thank you for the valuable comment. To address this issue, we have made the following corrections:

(i) We have corrected the typo for count (N) for personality variables from 3456 to 96. 

(ii) Table 1 has been split into two separate tables: Table 1a and Table 1b presenting the descriptive statistics for poem-related and personality variables, respectively.

(iii) In the “Procedure” section of the revised manuscript, we have clarified the scale used for scoring personality traits as follows: “All personality questionnaires utilized a 7-point scale, with 1 representing “disagree strongly” and 7 representing “agree strongly”.

The separate tables, i.e., Table1a, and Table 1b are as follows: 

Table 1a. Descriptive statistics of the creativity and its potential predictors including mean, standard deviation, skewness, kurtosis, standard error (SE), and variance inflation factor (VIF).

Note: The VIF for a variable is defined for a set of predictor variables by 1/(1-R^2) where R^2 is for the model predicting the variable from all the other predictor variables.

Table 1b. Descriptive statistics of readers’ personality trait variables including mean, standard deviation, skewness, kurtosis, standard error (SE)

4. Major issue. Concerning figures 1-4 -all the figures with slopes. Please explain (also to the readers) what are the scales on X- and Y- axes. The original measurement scales were 7-point scales. On the figures we meet what seems to be re-centered scales, but they are not obvious, and in some cases the ranges are -6 to 4, which is strange. One would expect the ranges to be of size six. Explain your transformations.

Our response: Thank you for your valuable comments and suggestions. We have modified the scales of the interaction plots. This adjustment and the rationale behind the choice of scales have been mentioned in the “Analysis” section of the revised manuscript, as follows:

“The original measurement scales were based on 7-point scales. Before entering the model, five potential predictors were centered within each subject (i.e., group mean-centered) to obtain a clear estimate of the within-group effect (Enders, C. K., & Tofighi, 2007). For interaction plots, it is a standard practice to use a scale that reflects the original range of the variables rather than the centered range. Therefore, on the X-axis, the scales for the predictors (group mean-centered) vary from -7 to +7, while the outcome variable, which remains uncentered, is represented on the Y-axis with a range from 1 to 7.”

5. Lines463-465: "The current study investigated the moderating effects of four personality traits openness, intellect, awe-proneness, and curiosity - on the evaluation of creativity in English language poems: openness." 

Editor’s comment: What is 'openness' doing at the end of the sentence?

Our response: Thank you for pointing out this error. It has now been corrected. 

6. Lines 495-498: "The distinct moderating effects of openness on aesthetic appeal and felt valence in our study suggest that the aesthetic emotions evoked through affective processing of poetry might differ from the emotional valence experienced during the reading of poetry." 

Editor’s comment: I think I understand every word in this sentence, but I don't understand what it means. What is the difference between "affective processing of poetry" and "emotional valence experienced during the reading of poetry”? Please try to clarify this sentence, and the whole discussion section, stating things in very plain language, and taking in mind an audience who are not versed in your jargon.

Our response: Thank you for your valuable feedback. The part in the Discussion section has been clarified as follows: 

“Processing of any artwork, including literature, includes a component called “aesthetic emotion” (Chatterjee & Vartanian, 2014; Leder et al., 2004; Jacobs, 2015; Menninghaus et al., 2019). Aesthetic emotions are the discrete emotions that always include an aesthetic evaluation/appreciation and are further associated with subjectively felt pleasure or displeasure, i.e., felt valence, during any emotional episode (Menninghaus et al., 2019). Our study indicates that individuals with higher levels of openness may be less influenced by aesthetic emotions compared to those with lower levels of openness while assessing creativity of poems. On the flip side, higher open individuals seem to be more positively impacted by the overall aesthetic appeal of poems compared to those with lower levels of openness. This notion aligns with the understanding that aesthetic appeal appreciation and evaluation of artwork, beyond aesthetic emotions, involves processing of other inherent features of art, such as styles, experience of pleasure of generalization (Leder et al., 2004; Hartley & Homa, 1981; Gordon & Holyoak, 1983), and knowledge (Silvia, 2010; Lachapelle et al., 2003; Cupchik & László, 1992). ” 

The whole discussion section has been restructured for easy comprehension.

7 Line 525: “a finding corroborated by previous research (111)." Please reconsider your semantics. Your study might corroborate previous research but claiming that previous research corroborates your study is over the top.”

Our response. Thank you. This has now been rectified. 

8. Regarding lines 534-557: There can be an alternative 'explanation': individuals with high openness that are more motivated to learn, might know more, and thus be less surprised in general. They might also be less surprised by the contents and forms in the poems. This could be checked with your data: are the average surprise ratings of open individuals lower than surprise rating of not open individuals? Do low-openness individuals put more emphasis on surprise, or maybe they are just more surprised? Or both? (the two things are not the same).

Our response: 

Thank you. We appreciate your valuable comment. We specifically discussed the results by examining the simple slopes of the interaction effects involving the personality trait variables. As suggested, we revised the descriptions to include the slope differences for high and low openness as: 

“The statistically significant difference of the simple slopes for high and low open individuals indicates that, more open individuals, who are more motivated to learn, might be less influenced by the surprise in the contents of the poems compared to their lower counterpart while judging poetic creativity.The transient and ambiguous nature of surprise might disrupt their affective states, leading to a reduced impact of surprise on their creativity judgment. In contrast, less open individuals perceived surprise as a more significant factor in their evaluation of poetic creativity than their higher counterparts, contradicting our initial prediction. It is noteworthy that the interaction does not indicate that high openness readers were less surprised by the poems compared to low openness readers. Rather it suggests that their judgments of a poem’s creativity were less influenced by the surprise element of the poem compared to those with lower openness. Furthermore, our focus was not on whether individuals with higher openness rated surprise more highly on average than those with lower openness. Instead, we focused on the differential level of surprise ratings for high and low openness. Our objective was to investigate whether there was a difference in how surprise was prioritized as a predictor of creativity judgment between the two levels of openness.” 

We have incorporated the results of the statistical test for the slope difference (highlighted in green) into the Table 6 in the revised manuscript, as follows:

Table 6. Results of simple slopes analyses for the high and low levels of the moderators and differences in slopes.

9. Major issue. The discussion section contains too much speculation, some of which could actually be checked with existing data. Perhaps in another paper? For this paper, please tone down the speculations. I will restate it in different terms: when reading the discussion, I did not feel awe, just confusion. One of the reviewers complained on the lack of conceptual clarity. I think it acutely applies to the discussion. Make the discussion clearer and less speculative. Another aspect of the discussion: it contains mostly abstract words, many of them nominalizations, and many key nouns are modified by adjectives. This makes reading more difficult. Try writing this section in shorter sentences, it might help. Try using less subordinated clauses.

Our response: Thank you. The discussion section has been revised thoroughly and restructured accordingly. 

10. Lines 554-557: "Our results also suggest that awe-prone individuals are more curious (120), and that awe itself can stimulate curiosity (121)." 

Editor’s comment: When one says "results suggest" it implies something new; but you have references, so it is old. Consider rephrasing.

Our response: Thank you for the helpful comment. We rephrased the line as: “our results indicate that awe-prone individuals are more curious and that awe itself can stimulate curiosity; these results are in line with previous research (Anderson et al., 2020; Izard, 1977).”

11. Lines590-591: "We, here operationalized ‘creativity’ as the poem’s creative potential." 

Editor’s comment: Regarding the issue, raised by a reviewer, on whether creativity is a property of people or of results. It is a difficult issue, but a notion of 'creative potential' is not likely be understood by the audience. Here is a suggestion, by twisting the 4P approach. Results can often be called 'creative', not by attributing them creativity per se, but just by implicitly denoting them as results of a creative process. Such implicit attribution is pervasive, so we call many things 'creative' without implying that the things can create something.

Our response: Thank you for your valuable suggestion. We removed the afore-mentioned lines from the limitations section. Instead, we integrated the explanation into the Introduction section, incorporating the 4P model as suggested. Here is the revised passage:

“The 4P model of creativity, a seminal theoretical framework of creativity, proposed “The word creativity is a noun naming the phenomenon in which a person communicates a new concept (which is the product). Mental activity (or mental process) is implicit in the definition and of course no one could conceive of a person living or operating in a vacuum, so the term press is also implicit. The definition begs the questions as to how new the concept must be and to whom it must be new” (Rhodes, 1961, p. 305). Among these 4P approaches, i.e., person, product, process, and press, the product or physical object, plays an important role. In common perceptions, creativity is often equated with its tangible outcome — the creative product. When asked to define creativity, many would instinctively describe it in terms of the final product (Gruszka et al. 2017). Literature suggests that a product-centered operational definition is the most useful for empirical research in creativity and presumably the most important feature of this definition is its reliance on subjective criteria (Amabile, 1982). Despite debates and the difficulty of precisely defining creativity of a product (Sternberg & Lubert, 1999; Beghetto, & Dow, 2004; Simonton, 2018), the most widely accepted operational definition is the “standard definition” of creativity (Runco & Jaeger, 2012), which states that for a product or idea to be deemed creative, it must be both original or novel and useful or appropriate. Additionally, surprise is also added as the third ingredient of creativity (Simonton 2012). The process aspect of the 4P model usually involves two phases of cognitive processes: the generative phase and the evaluative phase (Fink et al., 1992). 

The present study adopts a dual focus on both the product and process aspects of creativity using the poem as the product and its evaluation process as the measure of creativity. We operationalized the ‘creativity’ of a poem as its creative potential, aiming to broaden the understanding of creativity from the creator to the creation itself. Our approach is in line with past studies that have investigated the creativity evaluation of various types of products/artefacts, such as ideas (Lloyd-Cox et al., 2022), musical compositions (Pinho et al., 2014; Zioga et al., 2020), short stories (Toivainen et al., 2021), and product concepts (Guenther et al., 2021), to name a few. This approach allows us to investigate how individuals assess the creativity of poems, recognizing the subjective nature of such evaluations and how they may be influenced by individual personality traits. In summary, we aim to uncover how variations in reader personality may subtly influence the evaluation of a poem's creativity, thereby shaping an implicit model of evaluation.”

Responses to the comments of Reviewer 1:

Reviewer #1: All my comments have been answered to my satisfaction. Nevertheless, I believe that it would be in the authors' own interest to explain their definition of the term "creativity" in more detail (and to do so at the beginning of the paper). I believe that a large proportion of readers do not know in which tradition the authors use the term. To avoid losing these readers, it would be helpful to explain what the "4P model" is and how to define creativity "through the lens of the product". Honestly, I believe that the majority of readers - like me - will not understand a single word when they read this. I would also recommend that you invest a little time to correct typos. To give some examples: Line 91: "artwork..", line 273: "onpersonality", line 465: "openess", ..

Our response: Thank you for your valuable comments and suggestions. Following your advice, we have provided an explanation of the definition of "creativity" in the introduction, along with an elucidation of the "4P model" for readers outside the field of creativity research. We have also corrected any typographical errors that were identified. The inclusion of the 4P model in the “Introduction” is as follows:

“The 4P model of creativity, a seminal theoretical framework of creativity, proposed “The word creativity is a noun naming the phenomenon in which a person communicates a new concept (which is the product). Mental activity (or mental process) is implicit in the definition and of course no one could conceive of a person living or operating in a vacuum,

---

## [Editor Report · Decision Letter 2]

1 Jul 2024

Beyond the words: Exploring individual differences in the evaluation of poetic creativity

PONE-D-23-18669R2

Dear Dr. CHAUDHURI,

We’re pleased to inform you that your manuscript has been judged scientifically suitable for publication and will be formally accepted for publication once it meets all outstanding technical requirements.

Kind regards,

Michael Flor

Academic Editor

PLOS ONE

Additional Editor Comments (optional):

1.

Lines 296-297: "Additionally, our sample represented a diverse range of professions and ethnicities."

This statement is not supported by any data.

This is a very minor aspect in this manuscript,

but if you make such a statement, it needs to be supported.

What to do: either remove it, or add, in a footnote, some brief basic breakdown of professions and/or ethnicities.
---

## [Editor Report · Acceptance letter]

5 Jul 2024

PONE-D-23-18669R2 

PLOS ONE

Dear Dr. CHAUDHURI, 

I'm pleased to inform you that your manuscript has been deemed suitable for publication in PLOS ONE. Congratulations! Your manuscript is now being handed over to our production team.

Kind regards, 

on behalf of

Dr. Michael Flor 

Academic Editor

PLOS ONE